## RESEARCH ARTICLE

# Steroid 21-hydroxylase deficiency dysregulates essential molecular pathways of metabolism and energy provision

Irina Bacila[1], Lara Oberski[1,2,*], Nan Li[1,2], Karl-Heinz Storbeck[3], Vincent T. Cunliffe[2] and Nils Krone[1,‡]

## ABSTRACT

The prevalence of metabolic disease is increased in congenital adrenal hyperplasia (CAH) due to 21-hydroxylase deficiency. However, the underlying molecular mechanisms causing these problems are not fully understood. We aimed to elucidate the metabolic phenotype and conduct a transcriptomic analysis of a 21-hydroxylase-deficient zebrafish model, to unravel the molecular mechanisms underlying the metabolic pathophysiology of CAH. The morphology, anatomy and transcriptomic analysis of whole larvae, adult liver tissue from 18-month-old *cyp21a2*−/− zebrafish were compared to those of wild-type siblings. Our main phenotypical finding was that adult mutants were larger, with increased fat deposition compared to controls, in-keeping with the transcriptomic analysis showing the dysregulation of several biological processes involved in lipid metabolism. Importantly, we found that ATP synthesis and provision of energy precursors were included among the most significantly suppressed processes in both larvae and adult livers. We conclude that cortisol deficiency in *cyp21a2*−/− mutants causes growth and body fat abnormalities at adult stages, as well as transcriptomic dysregulation of metabolic processes, energy homeostasis and inflammatory responses in both larvae and adults. These findings reveal how GC deficiency in zebrafish contributes to the development of the metabolic comorbidities that are similar to those observed in patients with CAH.

KEY WORDS: Steroid 21-hydroxylase deficiency, Zebrafish, Adrenal disease, Metabolic dysregulations

## INTRODUCTION

Congenital adrenal hyperplasia (CAH) due to 21-hydroxylase deficiency (21OHD) is the most common metabolic cause of inherited glucocorticoid (GC) deficiency, occurring in about 1 in 15,000 live births (Claahsen-van der Grinten et al., 2021). Patients require lifelong treatment with synthetic GCs, often in supraphysiological doses. This commonly fails to mimic physiological cortisol secretion and leads to daily fluctuations between glucocorticoid overexposure and glucocorticoid deficiency. Synthetic GCs have been associated with a wide variety of side effects, leading to significant metabolic and cardiovascular comorbidities and limiting the duration and quality of life in treated patients (Bancos et al., 2015). Metabolic complications, such as obesity and insulin resistance, have been traditionally attributed to GC overexposure in CAH patients who frequently require supraphysiological GC doses to suppress androgen excess (Prete et al., 2021). Androgens also have a direct effect on lipid metabolism, being regulators of adipocyte differentiation, adiponectin secretion from adipose tissue, lipolysis, lipogenesis and insulin signalling (O'Reilly et al., 2014; Lopes et al., 2021). By contrast, the pathophysiological consequences of daily intermittent GC deficiency on glucose and fat metabolism remain unclear. Ethical reasons prevent clinical research from studying the long-term effects of GC-deficiency in human subjects, which warrants translational studies using animal models.

The gene *cyp21a2* codes 21-hydroxylase, an enzyme involved in steroidogenesis in zebrafish and humans, both in the mineralocorticoid pathway where it catalyses the conversion of progesterone to 11-deoxycorticosterone, and in the glucocorticoid pathway, catalysing the synthesis of 11-deoxycortisol from 17-hydroxyprogesterone. In humans, 21-hydroxylase deficiency due to mutations of the *CYP21A2* gene leads to CAH, which is characterised by gluco- and mineralocorticoid deficiency, as well as androgen excess.

In zebrafish (*Danio rerio*), 21-hydroxylase is encoded by the *cyp21a2* gene (Weger et al., 2016), located on chromosome 16. Our group has produced a *cyp21a2*−/− mutant zebrafish line using TALEN mutagenesis and established it as an *in vivo* model of 21OHD by demonstrating that mutant larvae had reduced cortisol concentrations, impaired visual background adaptation and systemic downregulation of GC-responsive genes (Eachus et al., 2017). Published evidence from zebrafish has shown the *in vivo* impact of GC deficiency on several metabolic pathways in the absence of GC treatment (Weger et al., 2016). Based on these findings, we hypothesized that GC deficiency may contribute to the development of the comorbidities prevalent in patients with CAH, through the dysregulation of different metabolic pathways. Better understanding of the pathophysiological mechanisms responsible for these metabolic dysregulations will lead to more effective strategies of prevention and treatment. We conducted a study that investigated the effects of cortisol deficiency on metabolism and gene transcription using a zebrafish model of 21-hydroxylase deficiency (*cyp21a2*−/−). On measuring whole body steroid concentrations, we found that, in contrast with human 21-hydroxylase deficiency, this model has low androstenedione (an androgen precursor raised in patients with CAH) and normal testosterone. This allowed us to focus on the metabolic effects exclusively caused by GC deficiency without interference of any changes to androgens. In this study, we used *cyp21a2*−/− zebrafish larvae and 18-month-old adults to obtain an assessment of the metabolic phenotypes associated with cortisol deficiency by comparing them to their wild-type (WT) siblings. We confirmed

[1]Department of Clinical Sciences, School of Medicine and Population Health, University of Sheffield, Sheffield S10 2RX, UK. [2]Department of Development, Regeneration and Neurophysiology, School of Biosciences, University of Sheffield, Firth Court, Western Bank, Sheffield S10 2TN, UK. [3]Department of Biochemistry, Stellenbosch University, Ryneveld Street Stellenbosch 7600, Matieland, South Africa. *Present Address: Department of Infectious Diseases, Faculty of Medicine, Imperial College, London, UK.

‡Author for correspondence (n.krone@sheffield.ac.uk)

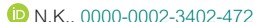 N.K., 0000-0002-3402-4727

cortisol deficiency by measuring full-body cortisol concentrations and the differential expression of cortisol response genes. We then conducted body measurements and dissections for the assessment of fat mass, using these outcomes as indicators of altered metabolism. Finally, we conducted a comprehensive gene expression analysis of larvae and adult livers using RNA sequencing. This was followed by transcriptomic analysis, to explore the molecular mechanisms involved in the development of metabolic abnormalities associated with GC deficiency. Comparative assessments revealed significant cortisol deficiency in *cyp21a2−/−* mutants, highlighted by the reduced expression of key regulatory genes such as *fkbp5*, and pronounced changes in overall body size, as well as the amount and distribution of adipose tissue. Additional sex-specific anatomical differences were observed, with female livers being larger than male livers in both WTs and *cyp21a2−/−* mutants. Transcriptomic analysis of both whole larvae and adult livers uncovered widespread dysregulations in metabolic processes, underlining the profound impact of 21-hydroxylase deficiency on metabolic health. These findings not only confirm the critical role of cortisol in maintaining metabolic homeostasis, but our data also highlight the *cyp21a2−/−* zebrafish mutant as a valuable model for studying the metabolic dysregulations associated with CAH. This investigation contributes to the fundamental knowledge required for developing more effective treatments for CAH, emphasizing the interplay between cortisol deficiency and metabolic health.

## RESULTS

### Whole-body steroid profiles of *cyp21a2−/−* adult mutants confirmed severe cortisol deficiency

We confirmed the validity of our *cyp21a2−/−* zebrafish mutants as a model for studying cortisol deficiency, by measuring whole body steroids in adult fish. Specifically, we measured cortisol, the steroid hormone precursor 17α-hydroxyprogesterone (17OHP), and androgens (androstenedione and testosterone), which are assessed in patients with 21 OHD to diagnose the disease and monitor treatment. The 21-hydroxylase deficient mutant adults had significantly lower cortisol concentrations than WT adults ($P<0.001$), confirming systemic GC deficiency. After excluding a WT outlier, 17OHP was significantly raised in mutants ($P<0.001$), similar to 21OHD in humans. By contrast, androstenedione was lower in *cyp21a2−/−* fish compared to WT siblings ($P=0.005$), the opposite of what is seen in CAH patients, while for testosterone and progesterone there was no difference between WT and mutant adults (Fig. 1A). Thus, our *cyp21a2−/−* zebrafish did not present with hyperandrogenism, which is a feature of 21-hydroxylase deficiency in humans. There were no significant differences between sexes for any of the steroid hormones measured.

Cortisol deficiency was also confirmed in *cyp21a2−/−* mutants by the downregulation of *fkbp5* in qPCR analysis, in both larvae and adult livers (Fig. 1B). Fkbp5 plays an important role in GR signalling (Gans et al., 2021), and *fkbp5* is an important GC responsive gene, its reduced expression indicating GC deficiency.

### Cortisol-deficient adult *cyp21a2−/−* fish are larger and have increased fat mass compared to WT siblings

Cortisol-deficient *cyp21a2−/−* mutant zebrafish were viable to adulthood and developed as both male and female adults, capable of reproduction. Weight and length measurements of adult fish showed *cyp21a2−/−* mutants to be both heavier and longer than WT (Fig. 1C). This feature was very consistent in males, being present in two different populations of fish for both weight ($P=0.003$ for the first and $P<0.0001$ for the second population) and length ($P=0.001$ for the first and

$P<0.0001$ for the second population). In females, the differences in weight and length were only significant in one population.

The secondary sexual characteristics, including the pigmentation of the fins and body, body shape and the prominence of the genital papilla, were no different between mutant and WT siblings from the same generation (Fig. 1D). WT adult male zebrafish exhibit strong stripes of golden pigment in the anal fin, interposed with the black stripes; this feature is absent in females. Female WT fish have a visible genital papilla anterior to the anal fin. These features were previously reported to be altered, as abnormal external sex characteristics, in other mutant zebrafish lines with androgen deficiency (Oakes et al., 2019).

Following the dissection of adult fish, we observed that *cyp21a2−/−* mutants had increased visceral fat deposition compared to WT, with more extensive fat deposits around the organs of the gastrointestinal tract. This feature was much more marked in males than in females. Moreover, all mutant fish, whether male or female, displayed an increased amount of fat surrounding the brain and subcutaneous fat compared to control siblings. (Fig. 1E).

Because glycemic control is an essential metabolic function, we wanted to know if there was an impact of cortisol deficiency on blood glucose concentration. Blood glucose concentrations varied widely in WT and mutants of both sexes. There was no significant difference in blood glucose concentration between *cyp21a2−/−* mutants and WT siblings (Fig. S1).

### Female livers were larger than male livers in both WT and mutant fish

Female zebrafish had significantly larger livers compared to males (Fig. S2A), a feature that to the best of our knowledge has not been described before in zebrafish. This sex-specific difference was found in both WT ($P=0.0001$) and mutant fish ($P=0.004$) with no significant difference between WT and mutants pertaining to the same sex. These differences were maintained when using the liver weight to body length ratio, as means to exclude body size as a potential confounder for organ size (Fig. S2B).

### The *cyp21a2−/−* transcriptome revealed extensive dysregulation of metabolic processes in both whole larvae and livers from adults

To understand the molecular mechanisms that led to the adult *cyp21a2−/−* zebrafish phenotype, we proceeded to the study of differential gene expression by RNA sequencing and comparative transcriptomic analysis of WT and *cyp21a2−/−* mutant whole larvae and adult livers. Through this approach, we were able to identify differentially expressed genes affected by cortisol deficiency and provided insights into disrupted biological pathways and molecular networks using comprehensive bioinformatics techniques.

The sequencing depth ranged between 14.2 and 18.2 million mapped reads for larvae (Table S1) and 32 and 48 million reads for adult livers (Table S2).

### Differential gene expression

Principal component analysis (PCA), plotting the first two principal component values of RNA-sequencing, showed clear distinction between WT and mutant larvae, also confirmed by the cluster dendrogram (Fig. S3A,B). Differential gene expression (DGE) analysis identified 398 genes that were significantly upregulated [log2 fold change (LFC) >0, $P$-adj<0.05] and 698 genes significantly downregulated (LFC<0, $P$-adj<0.05) in mutants. The 15 genes with most significantly dysregulated expression included cortisol response gene *fkbp5* (the most downregulated gene, LFC=−5.05,

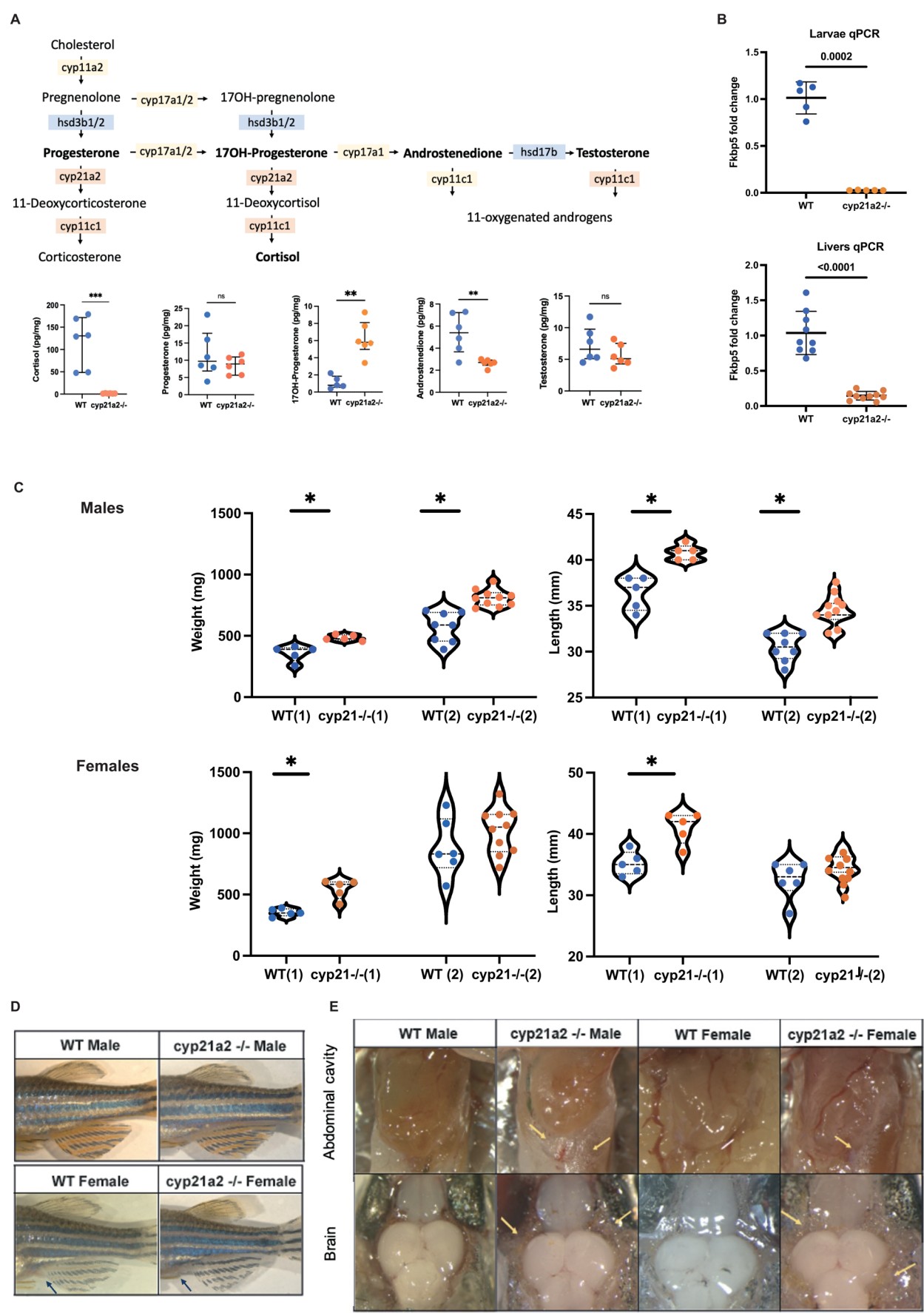

**Fig. 1.** See next page for legend.

**Fig. 1. Adult *cyp21a2*−/− mutant zebrafish phenotype.** (A) Diagram of intrrenal steroid biosynthesis in zebrafish (above) and whole-body steroid measurements in mutant fish (orange) compared to WT (blue). The horizontal lines represent median with interquartile range. *n*=6 fish per group. (B) Expression of *fkbp5* in larvae and adult livers. Comparison of whole-body steroids between WT (blue) and mutant (orange). (C) Weight and length measurements in *cyp21a2* male zebrafish, WT (blue) and mutant (orange). The numbers in brackets correspond to two different fish populations. (D) External phenotypes of *cyp21a2* mutant and WT sibling fish. (E) Dissection images. The upper pictures show the abdominal visceral fat flanking the gastrointestinal tract. The lower images were obtained on dissecting the skull, visualising the brain and the subcutaneous fat around it. The visceral and subcutaneous fat is indicated by yellow arrows (*indicates statistical significance).

*P*-adj=5.92e-300), and genes involved in glucose and protein metabolism: pyruvate dehydrogenase kinase 2b (*pdk2b*), and protein alkylation gene transglutaminase 1 (*tgm1l4*) (Fig. 2A).

In the study of differentially expressed genes in adult livers, PCA showed poor differentiation between mutants and WTs when both sexes were analysed together; however, there was clear difference between males and females (Fig. S3C,D), indicating that the differences in gene expression induced by sex override the effects of cortisol deficiency associated with the disrupted steroidogenesis caused by the mutation of the *cyp21a2* gene. Therefore, males and females were studied separately in the subsequent transcriptomic analysis. In male livers, a satisfactory level of clustering between WT and mutants was detected in the PCA analysis (Fig. S3E,F). DGE analysis identified 524 genes that were significantly upregulated (LFC>0, *P*<0.05) and 595 genes significantly downregulated (LFC<0) in male mutant livers (Fig. 3A,C), with *hsd20b2* being the most significantly downregulated gene (LFC=−4.24, *P*-adj=1.52e-28). In females, the DGE did not produce a clear distinction between mutants and WT livers (Fig. S3G,H). There were 541 genes significantly upregulated (log2 fold change (LFC>0, *P*<0.05) and 862 genes downregulated (LFC<0. *P*<0.05) (Fig. S3B,D). The most significantly downregulated identifiable gene was eukaryotic translation initiation factor 4E binding protein 3 (*eif4ebp3*) (LFC=−4.73, *P*-adj=8.42e-10) and upregulated, apolipoprotein A-IV b, tandem duplicate 2 (*apoa4b.2*) (LFC=2.08, *P*-adj=8.35e-9),

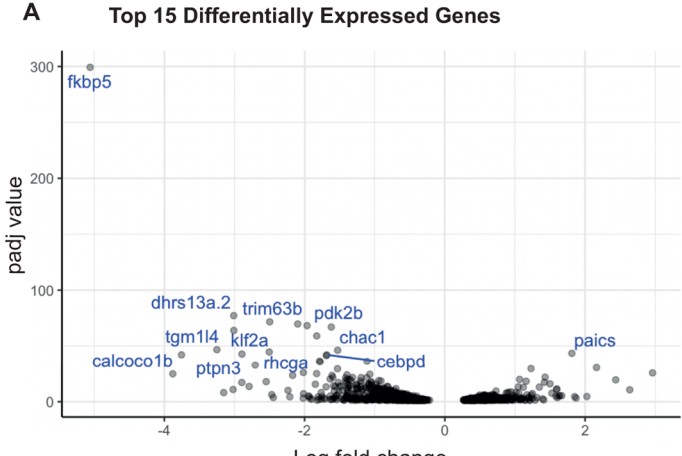

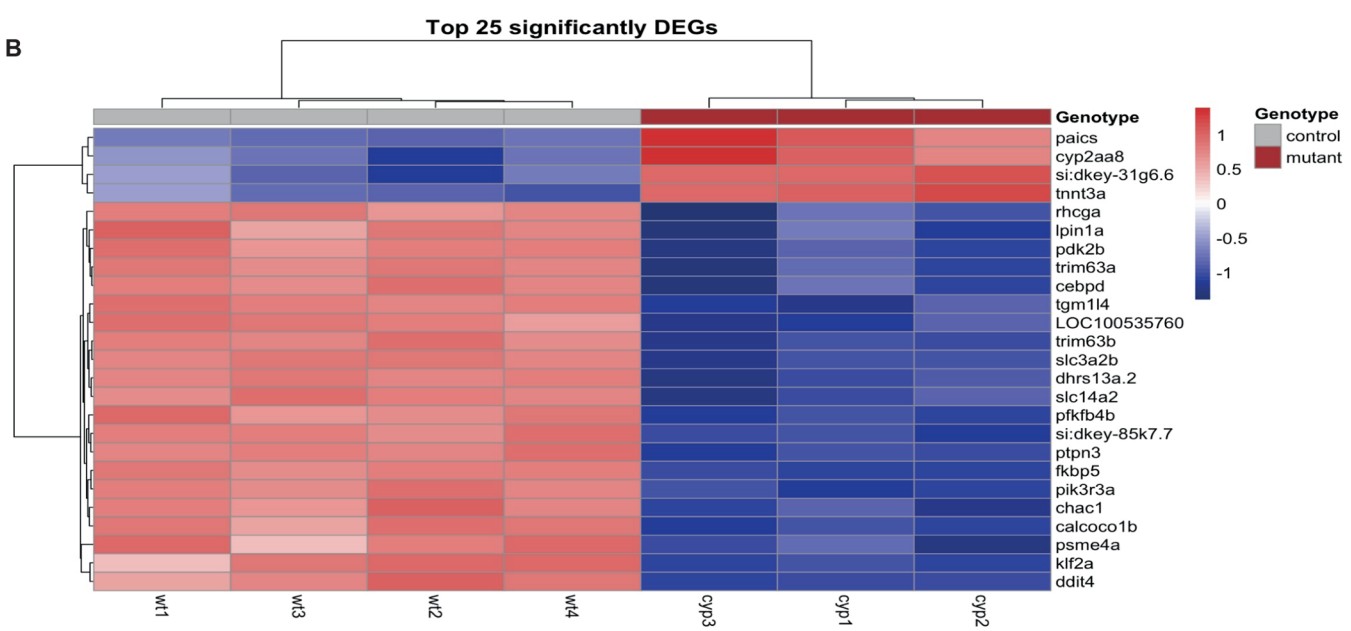

**Fig. 2. Differential gene expression in *cyp21a2*−/− larvae.** (A) Volcano plot showing the top 25 DEGs in *cyp21a2*−/− larvae, with statistical significance (*P*-adj) represented on the vertical axis and the LFC on the horizontal axis. (B) Heatmap and table indicating the top 25 DEGs in *cyp21a2*−/− larvae.

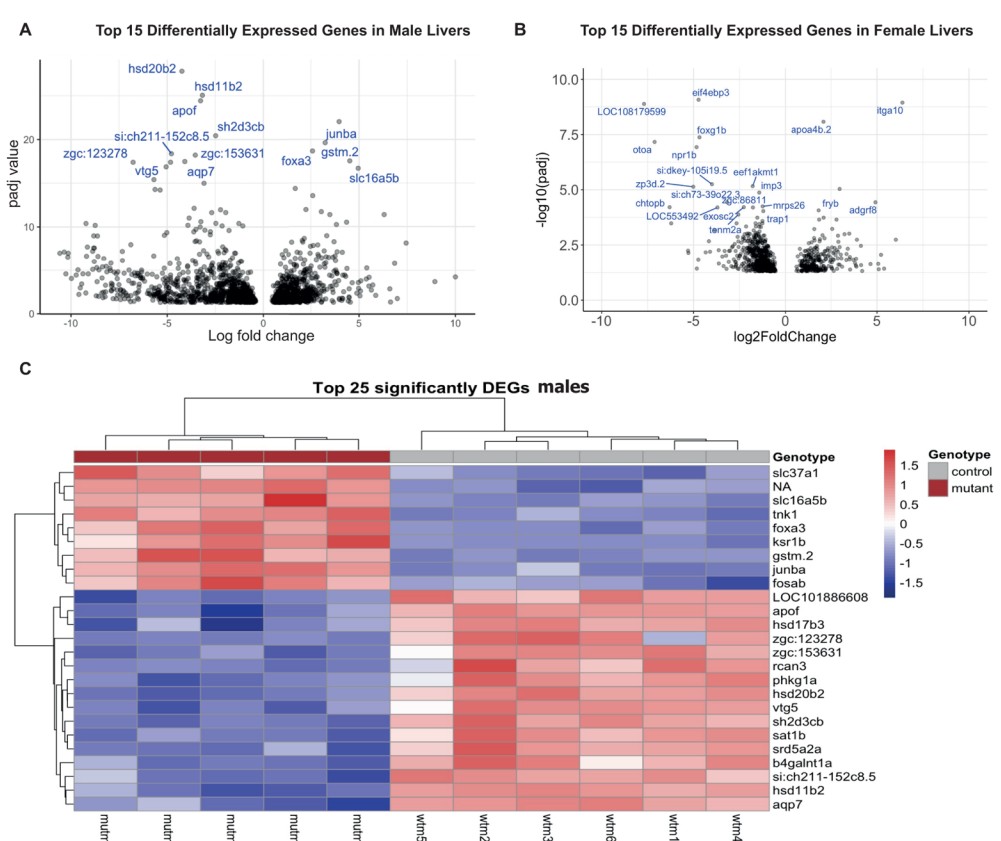

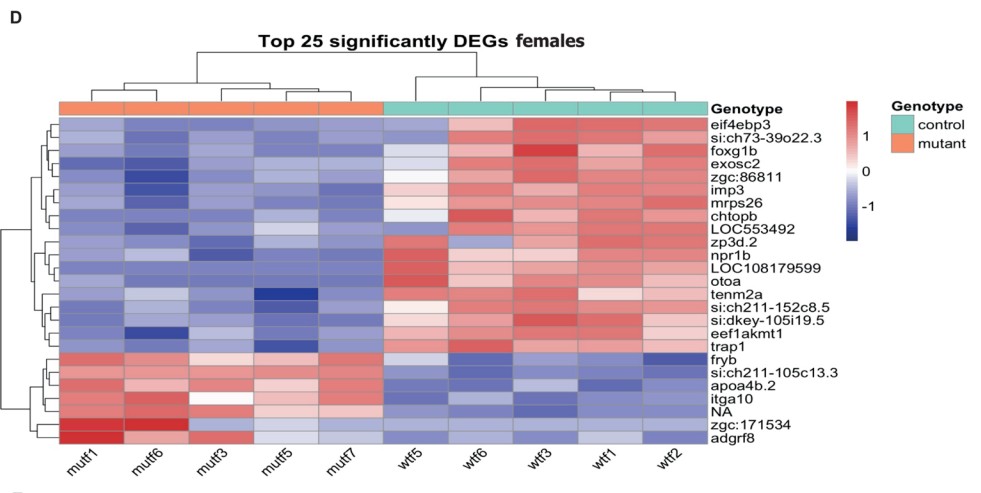

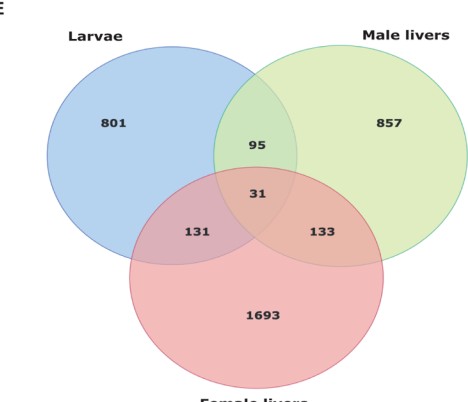

**Fig. 3. Differential gene expression in the *cyp21a2−/−* adult livers.**
(A,B) Volcano plot showing the top 25 DEGs in mutant livers in males (A) and females (B), with statistical significance (*P*-adj) represented on the vertical axis and the LFC on the horizontal one. (C,D) Heatmap indicating the top 25 DEGs in mutant livers in males (C) and females (D). (E) Venn diagram showing all the genes that were differentially expressed in mutant larvae (blue), adult male livers (green) and female livers (red), and their overlap between the three groups.

which codes a protein involved in lipid transport. There was poor overlap between suppressed and upregulated genes in mutant male and female livers, indicating further that cortisol deficiency has different impacts on liver gene expression between males and females. (Fig. 3E)

### Enrichment analysis identified multiple dysregulated biological processes in the cyp21a2–/– larvae and adult livers

We conducted gene ontology (GO) enrichment analysis to understand the impact of cortisol deficiency on different biological processes, paying special attention to those involved in metabolism. The GO technique is used to study biological pathways and processes that are significantly dysregulated in a set of genes provided, using a stringent cut-off ($10^{-5}$) for statistical significance. GO enrichment was performed using the GOrilla (GO enrichment analysis and visualisation). In the cyp21a2–/– larvae among the ten GO terms identified (Table S3), the majority pertained to metabolic processes. In adult male livers, GO enrichment analysis performed in GOrilla identified 12 GO terms; among the top ten results, four were processes related to the metabolism of steroids, lipids, organic hydroxy compounds, and small molecule metabolism (Table S4). In female livers, GO enrichment analysis identified 102 GO terms, among the top ten, eight processes were related to the metabolism of cellular nitrogen, nucleic acid and organic cyclic compounds (Table S5).

We then applied gene set enrichment analysis (GSEA) to the RNA-sequencing data. GSEA represents another way of identifying biological processes and molecular mechanisms that are either enriched or depleted in a given gene set. We complemented our GO overexpression and GSEA by STRING analysis to identify likely protein–protein interactions that are affected by GC deficiency. Based on the initial differential gene expression results, GO and GSEA analysis identified groups of genes involved in biological processes that were dysregulated by the cyp21a2 mutation; however, they do not offer information about the interactions between the proteins that are the products of these genes. The use of STRING provided an analysis of the likely physical interactions between proteins that are co-regulated in mutant zebrafish, which we perceived as an additional confirmation that hypocortisolism induces gene dysregulations that act synergistically at the molecular level, with an impact on whole processes and pathways.

Fig. 4A is a representation of the top 20 downregulated and upregulated biological processes found by the GSEA of cyp21a2–/– larvae. Among these, rRNA metabolism and ATP metabolism are two of the best represented among the biological processes suppressed in mutant larvae. The generation of precursor metabolites and energy was another significantly downregulated biological process (NES=−1.75, P-adj<0.001), with 56% of the dysregulated genes pertaining to it being also components of the ATP metabolism process. The GSEA results in male livers also showed significant downregulation of ATP metabolism, ATP synthesis, energy derivation, which echo the findings of GSEA in larvae. Processes that were upregulated included sterol synthesis and several biological processes involved in the immune/inflammatory response (Fig. 5A). Similar to males, in female livers, among the top 20 most suppressed biological processes were those related to energy homeostasis, including: the provision of energy precursors (NES=−1.8, P-adj<0.001), ATP metabolism (NES=−2.1, P-adj<0.001) and mitochondrial oxidative phosphorylation (NES=−2.4, P-adj<0.001) (Fig. 6A). However, we noted important differences between the GSEA in female and male livers. The marked dysregulation of the immune response seen in males was not present in females. Lipid homeostasis was

upregulated in cyp21a2–/– females (NES=1.8, P-adj<0.001), while lipid transport and localization were suppressed in cyp21a2–/– males. No other lipid metabolic processes were affected in females, while in males a large number of processes related to fat metabolism were dysregulated. Another significant distinction was the upregulation in female mutants of a large number of morphogenetic processes, which had not been identified in the male GSEA analysis (Fig. 6A).

We analysed the overlap of dysregulated biological processes (as identified by GSEA) between the different samples used for transcriptomic analysis: larvae, male and female adult livers, visualised by use of a Venn diagram (Fig. 7). We found that among the 16 biological processes shared between types of tissues, there were four suppressed processes pertaining to energy homeostasis: ATP metabolism, generation of precursor metabolites and energy, energy derivation by oxidation of organic compounds, and mitochondrial ATP synthesis coupled electron transport. There was better overlap in overexpressed processes between male and female adult liver samples, compared to the overlap between larvae and each type of adult samples.

### Dysregulation of ATP metabolism

We focused on ATP metabolism for the STRING analysis of the larval GSEA results, because it was found to be among the most strongly suppressed biological processes in cyp21a2–/– larvae and adult livers, and also due to the wide involvement of ATP in energy homeostasis and metabolism (Rigoulet et al., 2020). The STRING analysis in cyp21a2–/– larvae confirmed functional interactions between multiple proteins involved in different sub-processes of ATP metabolism, such as glycogen metabolism, the tricarboxylic acid cycle and the electron transport chain. Importantly, the proteins highlighted by the STRING analysis (Fig. 4B) corresponded to many of the genes that were significantly suppressed in the DGE analysis, including glycogenin1 (gyg1b, log2FC=−0.47, P<0.00001), glycogen synthase (gys1, log2FC=−0.69, P<0.00001), phosphorylase kinase (phkg1a, log2FC=−0.66, P=0.001), as well as other genes involved in the catabolism of carbohydrates: phosphofructokinase (pfkma, log2FC=−0.93, P<0.00001), aldolase a (aldoaa, log2FC=−0.39, P<0.00001), TP53 induced glycolysis regulatory phosphatase a (tigara, log2fc=−1.01, P=0.005). (Fig. 4B)

The number of proteins involved in ATP metabolism, and the interactions between them, were even more extensive in cyp21a2–/– male adult livers than in larvae (Fig. 5B). They pertained to multiple components of the mitochondrial oxidative phosphorylation system, the suppression of many important relevant genes highlighted by STRING being confirmed by the DGE analysis including: subunit of the mitochondrial respiratory complex NADH: ubiquinone oxidoreductase subunit A4 (ndufa4, log2FC=−4.75, P=0.0009), the cytochrome-c oxidoreductase complex, most significantly cytochrome c oxidase subunit 7A2a (cox7a2a, log2FC=−0.71, P=0.0001), as well as the tricarboxylic acid cycle. Additionally, several glycolytic enzymes were also downregulated, such as aldolase b, fructose-bisphosphate (aldob, log2FC=−1.69, P=0.032), as were different factors involved in glycogen metabolism such as the protein phosphatase 1, regulatory subunit 3Ca (ppp1r3ca, log2FC=−3.89, P<0.0001).

GSEA in female livers showed even more extensive dysregulations of APT metabolic processes compared to male livers and larvae; here, we found the highest number dysregulated genes coding of proteins involved in energy provision (Fig. 6B), as confirmed by DGE analysis: subunits of the mitochondrial respiratory complex: NADH: ubiquinone oxidoreductase complex assembly factor 1 (nudfaf1, log2FC=−1.1, P<0.001); cytochrome c oxidase assembly factor

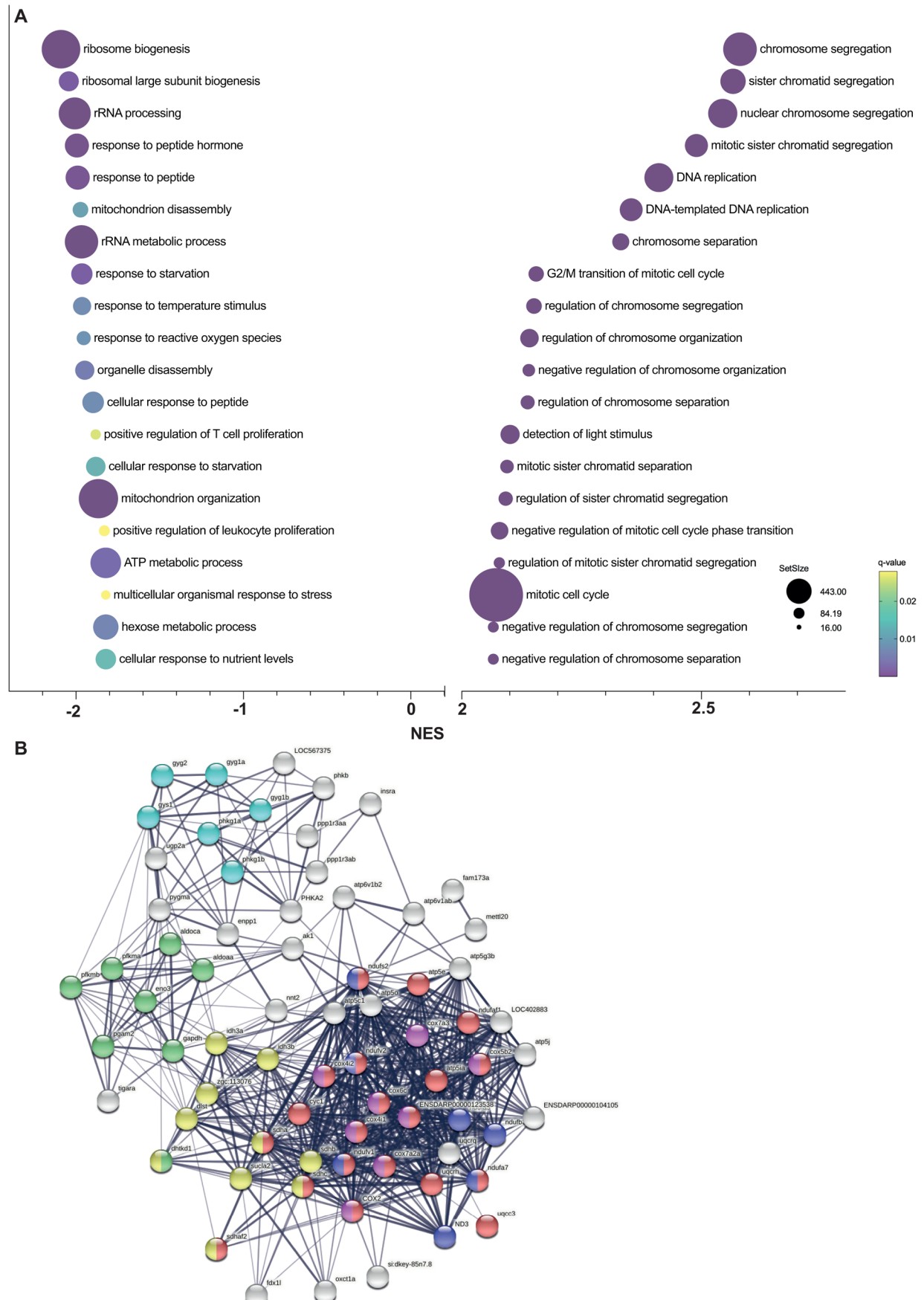

**Fig. 4.** See next page for legend.

**Fig. 4. Gene set enrichment analysis in *cyp21a2*−/− larvae.** (A) The top 20 dysregulated biological processes found by the GSEA. Bubbleplot produced in GraphPad Prism showing the 20 most downregulated (left) and upregulated (right) biological processes in *cyp21a2* larvae, based exclusively on the normalised enrichment score (NES). The size of the dots indicates the size of the gene set, and the colour corresponds to the q-value, decreasing from yellow to violet. (B) Visualisation in STRING of interactions between proteins encoded by genes involved in ATP metabolism whose expression is dysregulated in *cyp21a2*−/− mutant larvae. The connecting lines indicate interactions between proteins, the width of the lines being proportional to the confidence or the strength of data supporting the interaction. The colours of the nodes indicate the main sub-processes to which they belong: oxidative phosphorylation (red, number of proteins, *n*=20), NADH dehydrogenase (ubiquinone) activity (dark blue, *n*=7), cytochrome-c oxidase activity (pink, *n*=8), tricarboxylic acid cycle (yellow, *n*=10), glycogen metabolism (light blue, *n*=6) and glycolytic process (green, *n*=8).

COX14 (*cox14*, log2FC=−1.1, *P*=0.01), and the tricarboxylic cycle; several glycolytic enzymes were also downregulated, such as phosphoglycerate mutase 1a (*pgam1a*, log2FC=−1.1, *P*=0.002).

### Dysregulation of lipid metabolism in *cyp21a2-/-* adult male livers

In relation to our phenotypical findings in adult zebrafish mutants, in particular the increased fat deposition around organs, we also focused on the downregulation of lipid metabolism, which was much more prominent in male livers than in female livers and larvae. The main dysregulated processes included: beta-oxidation of fatty acids, as well as peptide biosynthesis, lipid, protein and amino-acid catabolism. These GSEA findings were confirmed by DGE analysis, which identified many genes involved in fat metabolism processes as being significantly suppressed. Among these, we highlight phospholipase D1b (*pld1b*, log2FC=−2.44, *P*=0.003) for lipid catabolism, regulators of fatty acid beta-oxidation, acyl-CoA synthetase long chain family member 5 (*acsl5*, log2FC=−8.35, *P*<0.001) and carnitine palmitoyltransferase 1B (muscle) (*cpt1b*, log2FC=−2.91, *P*<0.001), among the most significantly downregulated genes. Downregulated genes involved in lipid localization included apolipoprotein A-IV a (*apoa4*, log2FC=−7.83, *P*=0.003), intestinal fatty acid binding protein 2 (*fabp2*, log2FC=−9.10, *P*<0.001), vitellogenin 5 (*vtg5*, log2FC=−4.80, *P*<0.001), fatty acid binding protein 1b, tandem duplicate 1 (*fabp1b.1*, log2FC=−4.80, *P*<0.001).

We also analysed sterol synthesis, one of the top processes upregulated in male mutant livers, due to its metabolic importance especially in relation to cholesterol synthesis. STRING analysis showed multiple proteins involved in this process connected through functional interactions and the DGE analysis results confirmed that many of the related genes were significantly upregulated: methylsterol monooxygenase 1 (*msmo1*, log2FC=1.49, *P*<0.001), cytochrome P450, family 51 (*cyp51*, log2FC=2.0, *P*<0.001); and cholesterol synthesis gene 3-hydroxy-3-methylglutaryl-CoA reductase a (*hmgcra*, log2FC=4.1, *P*<0.001) (Fig. 5C).

### Upregulation of the inflammatory response in *cyp21a2-/-* adult male livers

Another feature of the transcriptomic analysis that was present in male livers, but not in females and larvae, was the dysregulation of inflammation which dominated the top 20 upregulated biological processes. Because of the important pathophysiological links between metabolic dysregulations and inflammatory response, we took a closer look at the processes involved in the immune response. Analysis in STRING showed important interactions between proteins that are involved in several components of the complement system

and which were found to be overexpressed by the DGE analysis: (*c3a.2*, log2FC=1.28, *P*=0.014; *c3a.3*, log2FC=1.01, *P*=0.001; *c3a.3*, log2FC=1.22, *P*<0.001). The cytokine-mediated signalling pathway was also upregulated, with the most significant dysregulation found for interleukin 12 receptor, beta 2a, like (*il12rb2l*, log2FC=1.81, *P*=0.002), tumour necrosis factor receptor superfamily, member 1a (*tnfrsf1a*, log2FC=1.98, *P*<0.001). Other significantly upregulated genes involved in the inflammatory response were MAPK activated protein kinase 2a (*mapkapk2a*, log2FC=1.96, *P*<0.001) and other members of the MAPK signalling pathway. The negative regulator of the JAK-STAT signalling pathway, suppressor of cytokine signalling 1a (*socs1a*) was also downregulated (log2FC=−1.55, *P*=0.332) (Fig. 5D).

### Associations of the transcriptomic findings from *cyp21a2*−/− larvae and adult zebrafish livers with human pathology

Finally, we sought to understand the relevance of our findings from RNA-sequencing data of *cyp21a2*−/− zebrafish in relation to the impact of CAH on metabolism in human patients. Thus, we studied the links between orthologues of the differentially expressed genes we identified and human disease. This was achieved through bioinformatic analysis using Disgenet2R to generate a list of significant human disease-associated orthologues.

In the mutant larvae (Fig. 8A), the predominant pathologies associated with orthologues of DEGs were cancers, including neoplastic conditions of variable histology and localisation. However, the findings also included a number of metabolic and cardiovascular problems, such as diabetes, metabolic syndrome, arteriosclerosis and coronary heart disease.

In male livers (Fig. 8B) the predominant pathologies were cardiovascular diseases and metabolic diseases; however, inflammatory bowel disease and cancer were also among the most common human conditions linked to the list of differentially expressed genes. The Disgenet2R analysis of WT and *cyp21a2*−/− mutant female livers showed similar results as observed in males, where metabolic syndrome, non-alcoholic liver disease and obesity were the three most important associations with regards to the amplitude (ratio) of similarity (Table S6). Cardiovascular, inflammatory, and neoplastic disease were also significant associations in female livers. However, there were also clear distinctions between males and females, in particular a strong association with disorders of glucose metabolism, especially diabetes mellitus and related comorbidities, which was not found in males.

### DISCUSSION

In this study, we aimed to understand the biological processes and molecular mechanisms involved in the development of metabolic problems in patients with CAH, with a particular focus on the contribution of GC deficiency to their pathogenesis. Our adult viable zebrafish *cyp21a2*−/− mutant line provided a unique opportunity to explore the impact of GC deficiency in larvae and adults, in the absence of synthetic GC replacement. We identified the changes that GC-deficiency produced in the zebrafish transcriptome at larval stage and also in the adult liver. We further sought to establish a link between the liver transcriptome and aspects of the *cyp21a2*−/− adult mutant body phenotype.

### Increased body size and fat mass are accompanied by dysregulated metabolic pathways in 21-hydroxylase-deficient fish

Our findings indicate that *cyp21a2*−/− adult fish are fertile and display normal external secondary sexual characteristics. However,

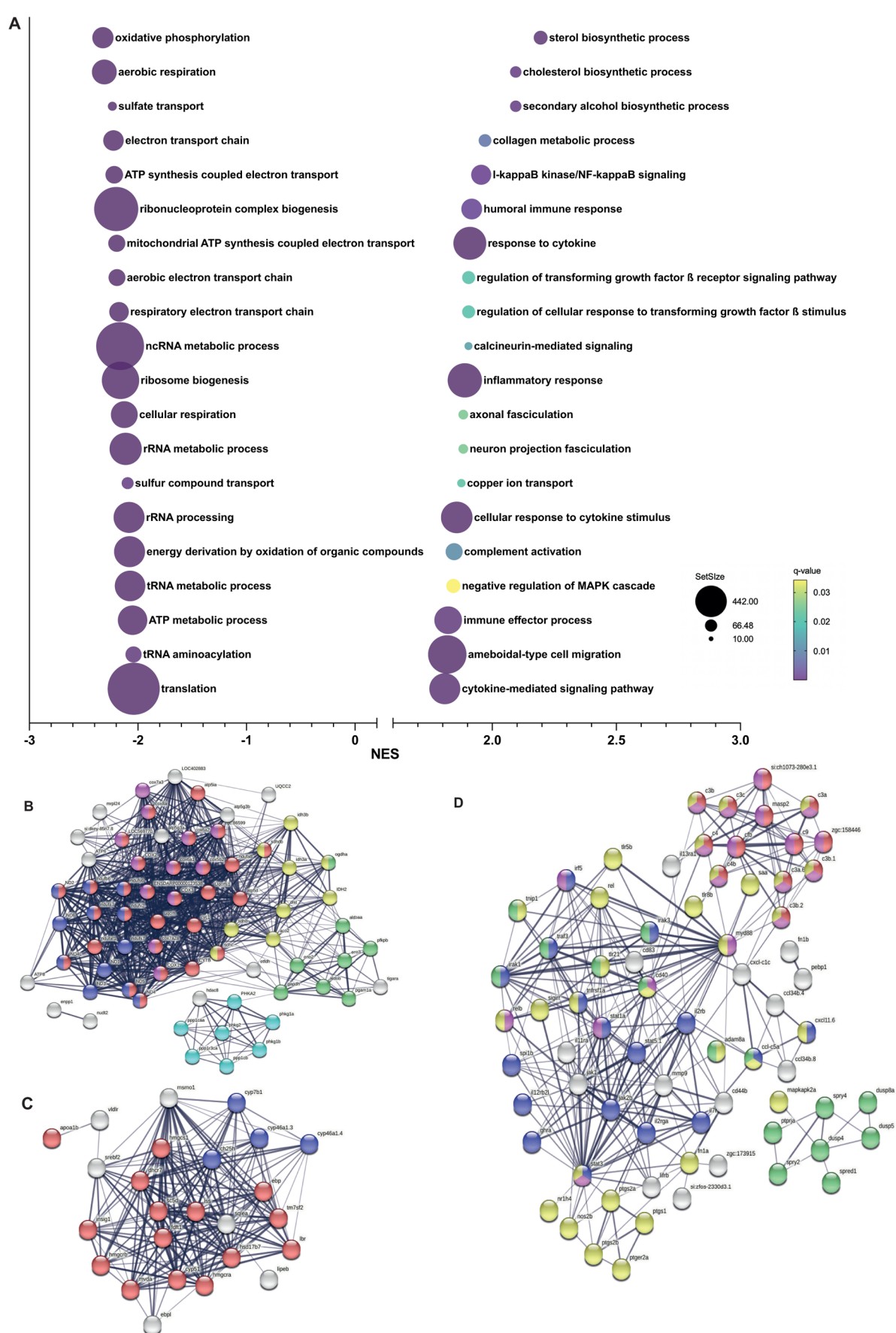

**Fig. 5.** See next page for legend.

**Fig. 5. Gene set enrichment analysis in *cyp21a2*−/− male livers.** (A) The top 20 dysregulated biological processes found by the GSEA. Bubbleplot produced GraphPad Prism showing the 20 most downregulated (left) and upregulated (right) biological processes in *cyp21a2* male livers, based exclusively on the NES. The size of the dots indicates the size of the gene set, and the colour corresponds to the q-value, decreasing from yellow to violet. (B-D) Visualisation in STRING of interactions between proteins encoded by genes involved in biological processes (colours of the nodes indicate the main sub-processes to which they belong). (B) ATP metabolism: all oxidative phosphorylation (red, *n*=29), NADH dehydrogenase (ubiquinone) activity (dark blue, *n*=12), cytochrome-c oxidase activity (pink, *n*=12), tricarboxylic acid cycle (yellow, *n*=9), glycogen metabolism (light blue, *n*=7) and glycolytic process (green, *n*=8). (C) Sterol synthesis (red, *n*=15), proteins with steroid hydroxylase activity (blue, *n*=4). (D) Immune response and MAPK signalling: immune effector process (pink, *n*=21), complement activation (red, *n*=13), inflammatory response (yellow, *n*=36), cytokine mediated signalling pathway (blue, *n*=18) and regulation of MAPK cascade (green, *n*=17).

both males and females are heavier and longer compared to WT siblings. This feature was more evident and more consistent in males, as in females weight fluctuates significantly in relation to the number of oocytes present in the ovaries. Thus, in future experiments a comparison of the difference between the weight of whole body and that of the ovaries may be a more relevant assessment option. Importantly, as a parallel to human disease, these resonate with the finding that patients with CAH have an increased prevalence of excessive weight even if not treated with excessive GC doses GCs (Bacila et al., 2022). Moreover, the macroscopic findings on dissection showed a tendency for increased visceral and subcutaneous fat in *cyp1a2*−/− mutant fish.

Our findings were unexpected, based on previous research using murine models of 21-hydroxylase deficiency. Previous studies using *Cyp21a2*−/− mouse models (Gotoh et al., 1988, 1994; Naiki et al., 2016; Thirumalasetty et al., 2024) reported decreased weight in mutants. Of note, one of the mutants (Gotoh et al., 1988; Gotoh et al., 1994; Naiki et al., 2016) required daily glucocorticoid treatment from birth to survive, and for both models the measurements were conducted within the first 5-10 weeks of life. Our zebrafish model, while severely cortisol deficient is compatible with life in the absence of GC replacement and our study assessed the body size and weight at 18 months of life. Thus, it is difficult to speculate whether the distinction in findings relates to differences between species, mutations, or duration of exposure to the disease. We considered several mechanisms and pathways as a possible explanation of these findings. Firstly, there is potentially a direct effect of cortisol deficiency on metabolism, causing the raised body size in mutant fish, through the dysregulation of lipid processing and transport, leading to increased deposition of fat. This is consistent with similar observations in several other animal studies, and human studies, indicating impacts of cortisol deficiency on metabolism (Christiansen et al., 2007; Kumari et al., 2010; Maripuu et al., 2016). Moreover, the transcriptomic analysis in the present study revealed the dysregulation of many genes involved in fat metabolism, such as the apolipoprotein A-IV a (*apoa4*), and the fatty acid binding protein 2 (*fabp2*), and of several biological processes involved in energy homeostasis, which we discuss further below. Increased body size was also found in other cortisol-deficient zebrafish lines that were established in our laboratory, including *cyp11a2*−/− (Li et al., 2020), *cyp11c1*−/− (Oakes et al., 2020), *fdx1b*−/− (Oakes et al., 2019), further supporting the proposition that glucocorticoids may act as regulators of anabolic and catabolic processes to maintain energy homeostasis, with impacts on energy stores and consequently fat deposition. While some of these lines

(*cyp11a2*−/−, *cyp11c1*−/−, *fdx1b*−/−) are also androgen-deficient, it is likely that GC deficiency is the key driver of metabolic dysregulation, given the wide involvement of GC in metabolism. Notably, in our transcriptomic analysis we found an overlap in overexpressed biological ribosomal, ribonucleoprotein, and RNA-related processes between *cyp21a2*−/− larvae, male, and female liver. Thus, we must consider the possibility that these dysregulations may also contribute to the growth and weight phenotypes of our mutant fish. Studies using murine models have demonstrated an effect of GC deficiency on the regulation of food intake and satiety (Makimura et al., 2003; Uchoa et al., 2009, 2012). Thus, GC deficiency was shown to be associated with an impaired satiety response and abnormal circadian feeding pattern that can result in increased food intake. However, previous evidence from zebrafish has demonstrated reduced food intake in response to stress, associated with overexpression of *pomc* (Cortés et al., 2018). As expected, *cyp21a2*−/− mutant larvae showed an increased expression of *pomc* in response to cortisol deficiency (Eachus et al., 2017), and it is important to recognize that POMC has anorexigenic effects (Uchoa et al., 2009). Taken together, this evidence suggests that the increased body size of our cortisol deficient *cyp21a2*−/− mutants is unlikely to be a result of altered steroidogenesis increasing feeding behaviours in zebrafish. Nevertheless, the mechanisms involved in appetite and satiety control in zebrafish are complex and further research is needed to understand the regulation of food intake and metabolism in cortisol-deficient zebrafish.

Interestingly, we did not identify differences in the blood glucose measurements between WT and mutant adult fish. However, the blood collection for glucose measurement, intended to accommodate dissections and tissue collection, was not conducted under fasting conditions. The fish were fed at 8:00 h and the blood was sampled between 9:00 h and 12:00 h, which may explain the variability of the glucose measurements and erase the differences between WT and mutant fish.

## Metabolic dysregulations induced by cortisol deficiency

We used transcriptomic analysis to gain a better understanding into the mechanisms through which GC deficiency contributes to the metabolic phenotype observed in the adult *cyp21a2*−/− fish. The focus on the liver in adult fish was based on its well-known wide involvement in regulating metabolism. We also compared the transcriptomes of WT and *cyp21a2*−/− mutant larvae to help understand the impact of cortisol deficiency on metabolic genes at different stages of development. Both the GO and GSEA methods that we used for the gene ontology over-representation analysis placed metabolism among the most significantly dysregulated biological processes in whole larvae and in adult livers. Small molecule metabolism was the most significantly affected metabolic process in both types of GO analysis. Small molecule metabolites are substrates, intermediates and products of metabolic processes (Hu et al., 2020) and the DEGs identified in *cyp21a2*−/− larvae pertaining to this GO term were relatively evenly distributed among lipid, carbohydrate and amino acid metabolism, suggesting a wide metabolic impact of cortisol deficiency affecting multiple interrelated pathways.

## Cortisol deficiency causes reduced biosynthesis of energy precursors

A most significant and consistent finding was the downregulation of genes involved in mitochondrial organization, ATP biosynthesis and provision of energy precursors. This finding was one of the most prominent consequences of cortisol deficiency in larvae, and it

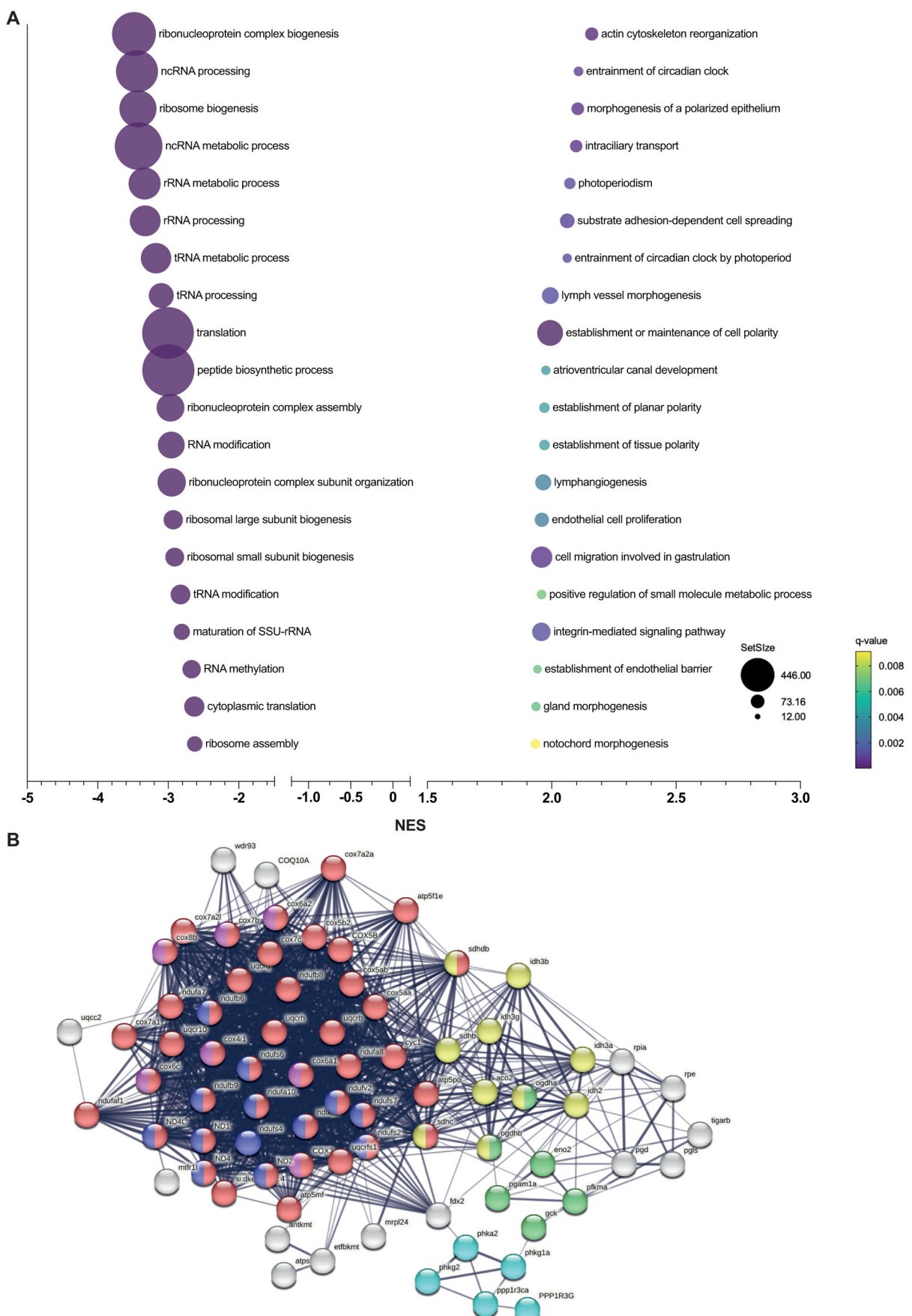

**Fig. 6.** See next page for legend.

**Fig. 6. Gene set enrichment analysis in *cyp21a2*−/− female livers.**
(A) The top 20 dysregulated biological processes found by the GSEA.
Bubbleplot produced GraphPad Prism showing the 20 most downregulated
(left) and upregulated (right) biological processes in *cyp21a2* female livers,
based exclusively on the NES. The size of the dots indicates the size of the
gene set, and the colour corresponds to the q-value, decreasing from yellow
to violet. (B) Visualisation in STRING of interactions between proteins
encoded by genes involved in ATP metabolism whose expression is
dysregulated in *cyp21a2*−/− adult female livers. The colours of the nodes
indicate the main sub-processes to which they belong: all oxidative
phosphorylation (red, *n*=44), NADH dehydrogenase (ubiquinone) activity
(dark blue, *n*=13), cytochrome-c oxidase activity (pink, *n*=7), tricarboxylic
acid cycle (yellow, *n*=10), glycogen metabolism (light blue, *n*=5) and
glycolytic process (green, *n*=6).

was also observed in adult livers, where it was even more marked.
This result is in keeping with the major role of GCs in energy
homeostasis and the use of energy reserves. This impact is also
indicated by the fact that several aspects of lipid metabolism,
including lipid transport and localisation, were downregulated
in the adult livers of the cortisol-deficient fish in this study.
The transcriptomic changes we observed in the *cyp21a2* mutant
zebrafish can explain, at least in part, the increased body size and fat
deposition phenotype we observed in these fish.

Due to the central role played by mitochondria in energy
homeostasis, extensive dysregulations of mitochondrial functions at
larval or earlier stages of development, are likely to have long term
maladaptive metabolic impacts. Interrelations between the human
stress axis and the functions of mitochondria have been explored by a
growing body of research, with a particular focus on defining the
effects of psychosocial stress (Picard and McEwen, 2018). It is well-
established that GC are involved in the regulation of mitochondrial
function due to the translocation of the GR into mitochondria (Du
et al., 2009). The impact of acute and chronic stress, as well as the
effects of GC treatment on mitochondrial function, have been
explored extensively by animal and *in vitro* studies (Du et al., 2009;
Psarra and Sekeris, 2011; Hunter et al., 2016). In hepatocytes,
GC have been shown to regulate mitochondrial transcription
both directly through the mitochondrial GR and by inducing
nuclear transcription of genes encoding mitochondrial transcription
factors (Psarra and Sekeris, 2011). Our results showed that there was a
suppression of mitochondrial organisation that extended to various
functions, including respiratory chain assembly, oxidation-reduction,
and protein transport, which would compromise core mitochondrial
functions and translate into poor efficiency in generating energy and
defective programmed cell death in homozygous *cyp21a2*−/−
mutants.

A significant impact of cortisol deficiency found in *cyp21a2*−/−
mutant larvae was the downregulation of ribosome biogenesis and
rRNA processing. Ribosomopathies have been described in the past
mostly in relation to the increased risk of oncogenesis (Farley-Barnes
et al., 2019). However, they can associate a number of pathologies,
including cardiac defects, facial malformations, hearing loss, liver
cirrhosis, neurological impairment. We did not identify any
phenotypical features to suggest similar effects in our *cyp21a2*−/−
fish. Regarding the metabolic effects of impaired ribosomal function,
data from Rpl11-deficient zebrafish and RPS19-deficient mouse
showed a suppression of the enzymes involved in glycolysis and the
biosynthesis of lipids and proteins, while the expression of genes
involved in catabolism is upregulated (Danilova and Gazda, 2015).
This would suggest that the downregulation of ribosome biogenesis
identified by our transcriptomic analysis may have a role in the
metabolic dysregulations we found in the adult fish livers.

## Dysregulation of lipid metabolism is associated with an upregulated inflammatory response in adult male livers

While lipid metabolic pathways were poorly represented in
the larval transcriptome, in mutant livers there was marked
downregulation of the beta-oxidation of fatty acids (FA). There
was also suppression of the mitochondrial organization that may
also have an impact on fatty acid metabolism (Houten et al., 2016).
Moreover, genes involved in lipid transport and localization,
including fatty acids (*cd36*, *fabp2*, *fabp1b.1*) and cholesterol
(*osbpl3a*, *anxa2b*) were also severely downregulated, indicating
an overall imbalance of lipid management within the liver. The
proinflammatory effects of cortisol deficiency (Coutinho and
Chapman, 2011) were apparent in *cyp21a2*−/− mutant adult livers
as the marked upregulation of several biological processes involved
in the inflammatory response, including the complement cascade
and cytokine signalling pathways. As in mammals, the roles of
cytokines and the complement system in humoral immunity have
also been previously described in zebrafish (Lieschke, 2001; Zhang
and Cui, 2014). In the adult livers of male *cyp21a2*−/− fish, there
was upregulation of the MAPK signalling pathway, which in
humans contributes to the development of arteriosclerosis and
hepatic steatosis in the context of reduced beta-oxidation of fatty
acids and abnormal lipid accumulation in tissues (Friedman et al.,
2018). Thus, a parallel can be drawn between our transcriptomic
findings on zebrafish livers and the mechanisms described in
the pathogenesis of human metabolic dysfunction-associated
steatotic liver disease (MASLD), previously known as non-
alcoholic fatty liver disease (Barbier-Torres et al., 2020). This
possible pathophysiological link is supported by the fact that
MASLD, steatohepatitis and even hepatic cirrhosis are among
the human conditions associated with our liver DGE gene list from
*cyp21a2*−/− adult fish livers. Cardio-vascular disease also ranked
high on the list of associated human diseases, and it is important to
note that inflammation and abnormal lipid homeostasis are major
determinants of the development of arteriosclerosis (Golia et al.,
2014; Barbier-Torres et al., 2020).

## Sex-specific effects of cortisol deficiency on liver metabolism

Interestingly, significant sex differences in the adult liver size and
transcriptome were found that overrode genotype distinctions and
have not been previously described in zebrafish. Research in the field
of neuroscience has extensively studied sex differences in the
functions and transcriptome of zebrafish brain (Zhai et al., 2022).
Our findings suggest that the liver metabolic processes involved in the
management of food and in energy homeostasis are different between
WT adult male and female zebrafish. In mammals, including humans,
where sex differences in energy metabolism have been suggested to be
an evolutionary adaptation that led to females becoming much better
equipped to resist food shortages, and thus conserve energy to
facilitate successful conception and gestation during pregnancy
(Mauvais-Jarvis, 2024). This difference has resulted in female
animal models of metabolic disease often being excluded from
experiments, having a less severe phenotype than males (Mauvais-
Jarvis et al., 2017). Such an exclusionary approach has been lately
discouraged, as it is perceived that sex differences in disease
susceptibility are highly relevant to disease pathogenesis (Mauvais-
Jarvis et al., 2017). Another feature assigned to phylogenetic
development is the special role of the liver in female mammals
(Della Torre and Maggi, 2017). The liver is considered to possess
'metabolic flexibility', allowing it to become to an extent part of the
reproductive system, due to its functions in the production and

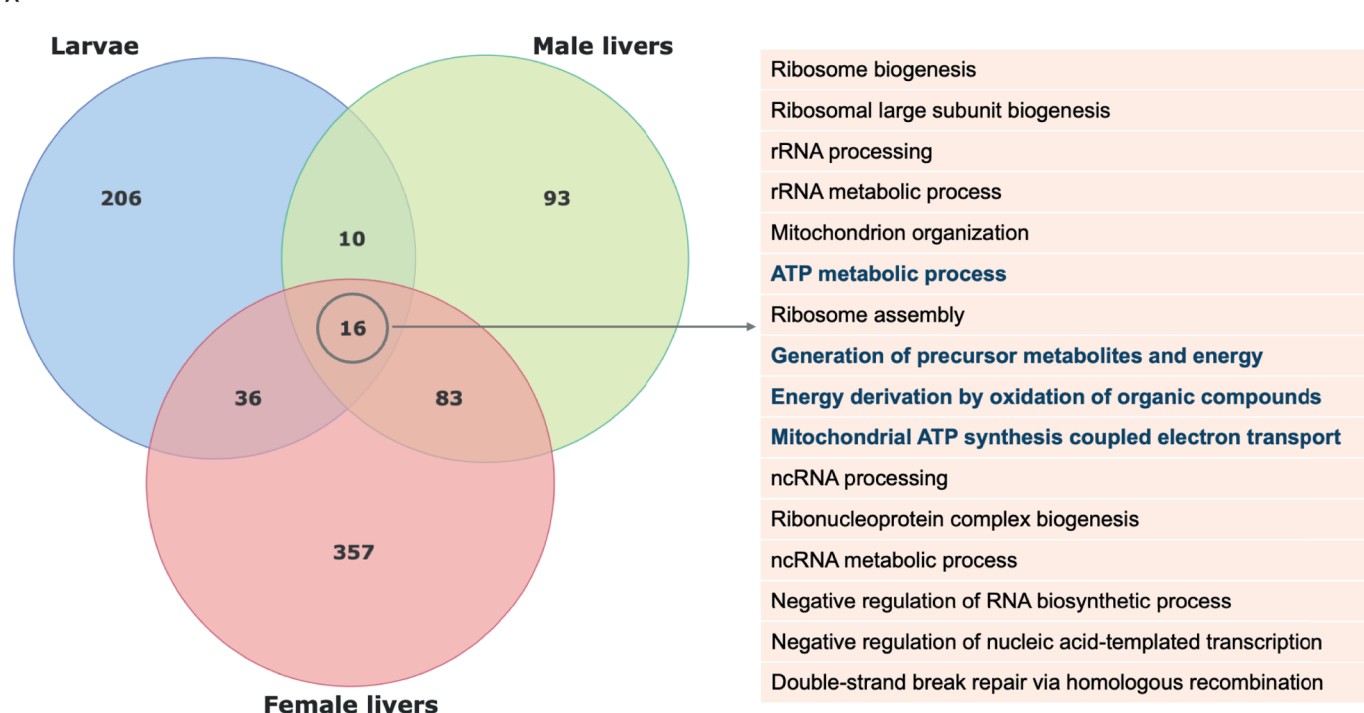

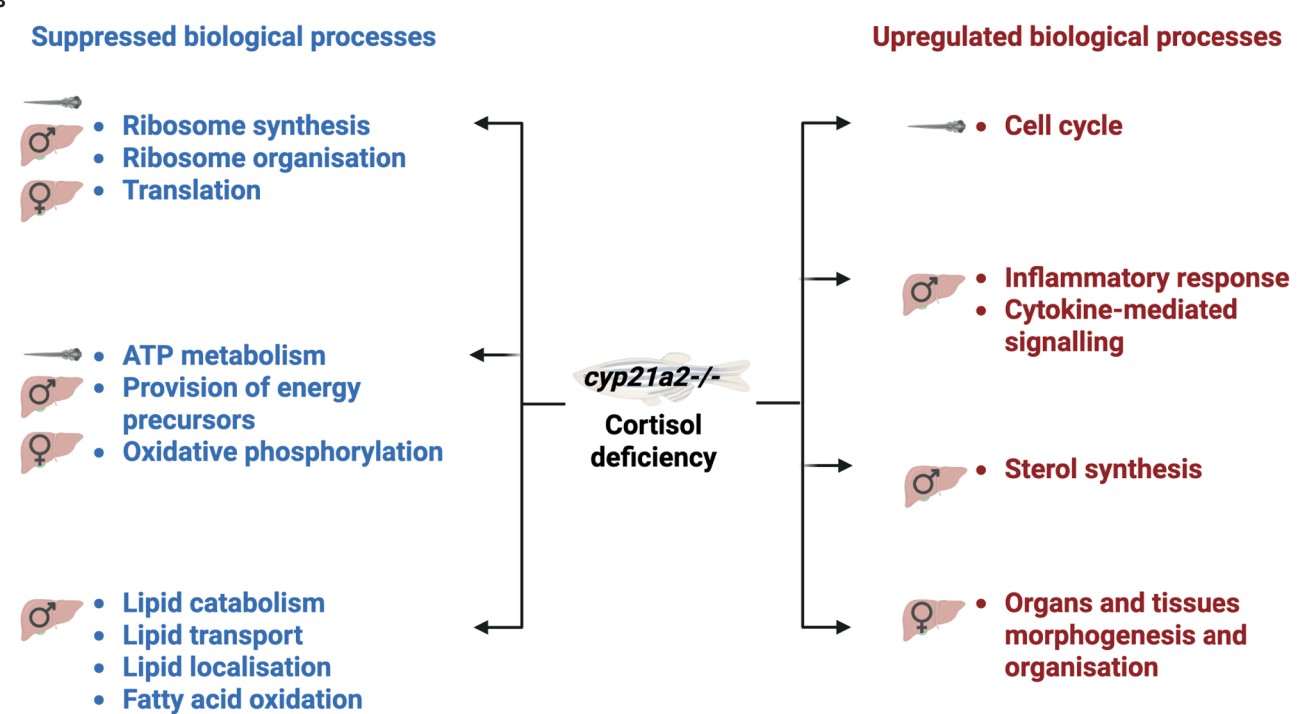

**Fig. 7. Summary of the GSEA results in the *cyp21a2−/−* zebrafish mutant.** (A) Overlap between the biological processes that were overexpressed in *cyp21a2−/−* mutant larvae and adult male livers. The upper Venn diagram shows all the genes that were differentially expressed in mutant larvae (blue) and adult male livers (red). The table on the right presents the 16 processes that were significantly dysregulated in all three groups. (B) The main dysregulations of biological processes as a results of cortisol deficiency in the *cyp21a2−/−* mutant larvae, male and female livers.

transport to the gonads of hormone precursors and nutrients necessary for reproduction and embryogenesis (Della Torre and Maggi, 2017). Applying this hypothesis to zebrafish could explain the observed differences between the zebrafish male and female liver transcriptome, specifically the upregulation in females of lipid transport and the vast number of upregulated biological processes related to reproduction and organ development. Thus, our findings suggest that this important evolutionary feature is conserved in zebrafish. The differences

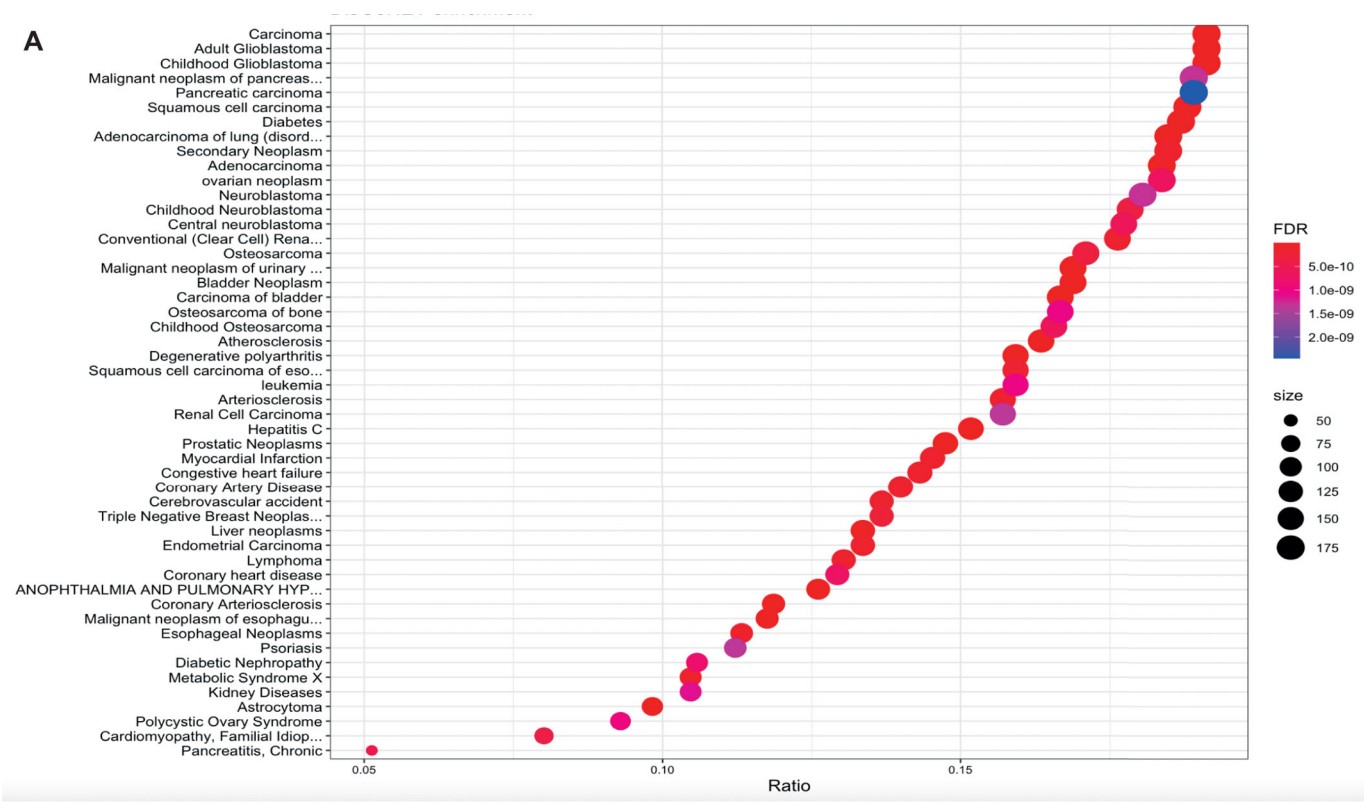

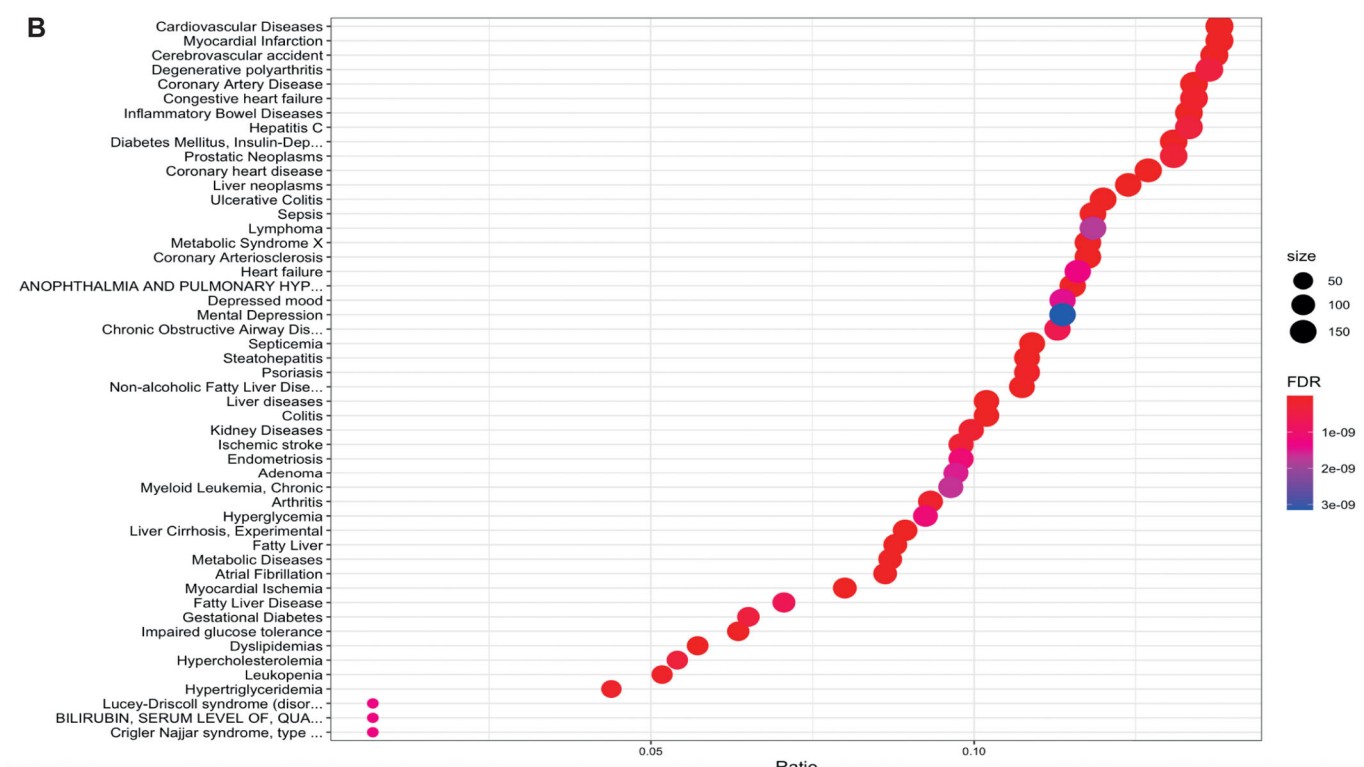

**Fig. 8. Associations between the *cyp21a2*−/− transcriptome and human disease in larvae (A) and male adult livers (B).** Enrichment plot showing the associations between the transcriptomic changes in *cyp21a2*−/− mutants and human diseases, produced in R using the 'disgenet2r' package. The horizontal axis shows the ratio between the number of DEGs in a category and all DGEs, the size of the dots represents the number of DEGs in a category and the colour, the *P*-adj value.

between sexes in the expression of genes responsible for metabolic processes may be part of the adaptive mechanisms that allow cortisol-deficient female fish to be able to reproduce.

The zebrafish is a convenient and reliable animal model, known to have many similarities to humans in terms of the structure and function of the stress axis, and the existence of evolutionarily

conserved factors involved in steroid biosynthesis and other metabolic pathways. Certain aspects need to be considered when interpreting the results. An essential distinction between GC-deficient zebrafish and humans with 21OHD is the absence of hyperandrogenism in zebrafish, which may contribute to the metabolic problems developed in CAH. Data from CAH patients demonstrate that androgens influence the prevalence of metabolic complications (Torky et al., 2021). However, this difference between zebrafish and human 21-hydroxylase deficiency also constitutes a practical advantage, as it allows us to focus the analysis of our zebrafish model on the effects of isolated GC-deficiency on metabolism.

Overall, our results obtained from the *cyp21a2−/−* zebrafish mutant provide in-depth insights relevant to the better understanding of metabolic problems associated with 21OHD, complementing clinical studies. The increased body size and body fat compared to WT controls resonate with the finding that patients with CAH have increased rates of overweight and obesity even if not treated with excess glucocorticoids. Without cortisol replacement, *cyp21a2−/−* zebrafish showed extensive dysregulations of many metabolic processes in the transcriptome of larvae and adult livers. The marked downregulation of genes involved in mitochondrial organization, biosynthesis of ATP and provision of energy derivatives was the most significant finding at both stages of development. The suppression of these processes by cortisol deficiency is in keeping with the interplay between stress axis, which is functional in zebrafish at 5 dpf, and mitochondria, on which cortisol acts via GR signalling. Our descriptive work of the phenotype and transcriptome of the *cyp21a2−/−* zebrafish mutants warrant further research to complete our findings through the quantification of the whole-body fat mass and histological analysis of liver fat deposition. The hypothesis emerging from our GSEA analysis is that cortisol deficiency impairs mitochondrial functions, including ATP biosynthesis, with a downstream effect on several metabolic processes that have ATP-dependent steps, such as fatty acid metabolism, lipid transport, protein biosynthesis and proteolysis. Moving forward, we will aim to test this hypothesis by exploring the density, structure, and function of mitochondrial complexes, and to further clarify the role of exogenous steroids in counteracting these mechanisms. Moreover, our work highlighted marked sex differences in the *cyp21a2*-dependent and -independent expression of genes related to liver metabolism, which could potentially represent an important evolutionary feature, and will require future research work for further clarification.

## MATERIALS AND METHODS
### Zebrafish husbandry
All procedures involving experimental animals were performed under the authority of licenses granted by the UK Home Office, in compliance with local and national animal welfare guidelines and in accordance with the UK Animals (Scientific Procedures) Act 1986. The zebrafish (*D. rerio*) were housed in the aquaria of the Zebrafish Facility of the MRC Centre for Developmental and Biomedical Genetics, University of Sheffield. Adult zebrafish were maintained in a recirculating system at 28.5°C on a 10-h dark: 14-h light photoperiod. Embryos were obtained by natural spawning, following in crossing of *cyp21a2−/−* or WT siblings, and incubated in E3 medium (containing 5 mmol/l NaCl, 0.17 mmol/l KCl, 0.33 mmol/l CaCl2, 0.33 mmol/l MgSO4) containing 2 µg/ml gentamicin, at 28.5°C. Developmental stages were assessed based on hours post fertilization (hpf) and days post fertilizations (dpf). No sex-ratio bias was noted in the *cyp21a2−/−* mutants.

### Genotyping *cyp21a2-/-* mutants
The mutant zebrafish line used in the experiments had been previously generated by our group using the TALEN approach (Eachus et al., 2017), causing a 14-bp deletion (c.del 211-224) in exon 2, leading to a frameshift with a premature stop at amino acid 96 (p.P70 fs26X). Mutant *cyp21a2−/−*

fish and WT siblings were obtained by in-crossing heterozygotes carrying the mutation and the progeny underwent genotyping following fin clipping biopsy, conducted by the aquarium technician. Due to the long time that had elapsed between the initial genotyping and the time adult fish were collected for experiments (18 months), consideration was given to potential inter-tank contamination. Thus, on culling the fish, a small portion of the caudal fin was collected to confirm the genotype. The material was placed in 100 µl NaOH (50 mM) and genomic DNA was isolated by heating at 98°C for 10 min, then cooling and adding 10 µl Tris pH 8. End-point PCR amplification of the targeted genomic region was carried out in a 20 µl reaction with 0.5 µl (10 µM) of each primer (forward: CTCTCGTGGGCTAAACAAGC and reverse: ACATGTATCCACCATTTGCG) and 1 µl genomic DNA in FIREPol® Master Mix (Solis Biodyne, Tartu, Estonia). The PCR program consisted of an initial activation at 94°C (2 min) followed by 36 cycles of denaturation at 94°C (30 s), annealing at 58°C (30 s), elongation at 72°C (30 s), and finally elongation at 72°C for 10 min. The PCR product (10 µl) was digested by adding 2 µl mix made of BseYI (0.25 µl), 10X buffer 3.1 (1.2 µl) and water (0.55 µl). The digests were analysed in comparison to the non-digested PCR product on a 1% agarose gel, with the mutation coinciding with the absence of cleavage of the 179-bp product into 103-bp and 76-bp products (Fig. S4).

### Morphological analysis and blood sugar measurement
Adult fish were humanely euthanised, their length and weight were measured and recorded, then they were photographed under the dissecting microscope. All fish were culled and dissected in the morning between 9:00 h and 12:00 h, having been fed at 8:00 h. Following tail ablation one drop of blood (0.5 µl) was expressed and blood sugar measured using a True Metrix glucometer. Subsequently, fish were dissected to expose the internal organs, which were also photographed. Organs and tissues (including gonads, kidneys, muscle, brain, liver and gastro-intestinal tract) were collected and frozen at −80°C for RNA extraction. Due to the observation that livers were larger in females compared to males, the weight of frozen livers was measured as well.

### Steroid hormone measurement from adult fish
The whole bodies of humanely euthanized adult 180 dpf *cyp21a2−/−* and WT siblings (three males and three females each) were snap frozen on dry ice. The samples were homogenized using a Mikro-Dismembrator S (Sartorius, Gottingen, Germany) and then freeze-dried. Approximately 100 mg of the dried samples were transferred to a glass test tube and resuspended in 900 µl MilliQ water and 100 µl MilliQ water containing internal standard. The steroids were extracted twice using 3 ml methanol. Following centrifugation for 5 min at low speed the methanol fractions for each sample were pooled and dried under a stream of nitrogen at 45°C. The dried residue was resuspended in 150 µl 50% (v/v) methanol prior to analysis. Steroids were separated and quantified using an Acquity UPLC System (Waters, Milford, CT, USA) coupled to a Xevo TQ-S tandem mass spectrometer (Waters) as previously described (O'Reilly et al., 2017).

### RNA sequencing and transcriptomic analysis of zebrafish larvae and adult livers
RNA was extracted from zebrafish larvae and adult livers using the Qiagen RNAeasy Kit. Total larval RNA was obtained from pooled 5 dpf larvae (*n*=30 larvae per sample) resulted from genotyped homozygous fish (*cyp21a2−/−* mutants and WT siblings). Four WT and three mutant larval RNA samples were sequenced. Livers were harvested and RNA extracted from 18-month-old mutants and WT fish (six males and six females for each group). The quality of the RNA extracted was determined by a 2100 Bioanalyzer (Agilent Technologies). Larval RNA samples were processed in the Institute of Cancer and Genomic Sciences, University of Birmingham, library preparation being performed using NEBNext® Ultra™ II Directional mRNA Library Prep Kit for Illumina®, followed by high-output 150 cycles, paired-end sequencing on a NextSeq 500 system. Liver RNA samples were sent to the Deep Sequencing Facility at Technische Universität Dresden where mRNA-Isolation via poly-dT enrichment and strand-specific RNA-Seq library preparation was followed by paired-end sequencing using a NovaSeq 6000 S4 Reagent Kit v1.5 (200 cycles) four-lane XP mode.

The raw RNA sequencing data was pre-processed using the University of Sheffield's latest central High-Performance Computing (HPC) cluster

ShARC. The fastq.gz files that were produced by RNA sequencing were first subjected to quality control using FastQC, which was also applied at each step of the subsequent pre-processing sequence. Next, rRNAs was removed with SortmeRNA version 2.1 (Kopylova et al., 2012), then the files were trimmed with Trimmomatic version 0.39 (Bolger et al., 2014) (ILLUMINACLIP:Trimmomatic 0.39/adapters/TruSeq3- SE.fa:2:30:10 LEADING:3 TRAILING:3 SLIDINGWINDOW:4:15 MINLEN:36). The Phred score threshold set when running Trimmomatic was Phred 33. FastQC quality control was also conducted after this step, Individual quality control reports were combined with MultiQC. (Ewels et al., 2016). Finally, the mapping/alignment of the reads to the reference zebrafish genome (GRCz11) was performed with STAR (Dobin and Gingeras, 2015).

Overall, the quality of the sequencing data was high, based on the per base sequence quality and per sequence quality scores. (Figs S5 and S6). The initial problem of overexpressed adapter sequences was effectively resolved by trimming of the data using Trimmomatic. Two of the quality metrics were consistently failed, respectively the GC content and the sequence duplication level. The left skewing of the GC content can be related to high content of rRNA; however, the samples were subjected to removal of rRNA by SortmeRNA. Another explanation could be sample contamination (bacterial or fungal material); however, as the left skewing was consistent among all samples, it is more likely that the high GC content is related to the library preparation process, in particular the PCR amplification (Browne et al., 2020). High sequence duplication level may be similarly caused by the PCR amplification involved in the library preparation protocol; however, it can also be triggered by the difference between over- and under-expressed genes. It is generally recommended that sequence duplications should not be removed as it may reduce the power of the analysis (Parekh et al., 2016). Thus, the results of the quality checks by FastQC and MultiQC were interpreted as validating the sequencing data and allowing the progress to the next steps in data analysis.

The bioinformatic analysis was performed in R (version 4.2.0) using the RStudio (version 4.2.1). QuantMode GeneCounts was used for gene count quantification from the reads of RNA-seq data for each gene for each sample (Liao et al., 2018). Differential expression analysis was conducted using the 'DESeq2' package. The threshold of significance was set at the adjusted $P$-value<0.05. Principal component analysis and heat map production was generated in R studio version 4.2.1, using the 'plotPCA' and 'pheatmap' packages, as part of exploratory data analysis.

Statistical overrepresentation analysis of GO terms with DEGs was conducted using two different bioinformatics tools. First, GO enrichment analysis was performed using GOrilla using all genes identified by RNA sequencing entered as the input, ranked by differential expression. A threshold of $P<10^{-5}$ was applied in GOrilla and REVIGO was used for visualisation. Secondly, the 'clusterProfiler' Bioconductor package in R was used to analyze a list of significantly differentially expressed genes ($P$-adj<0.05) against all annotated genes, with a q-value <0.05 set to identify enriched biological processes. Gene set enrichment analysis was conducted in R using the 'gseGO' function of the 'clusterProfiler' package. The input gene set consisted of the ranked list of all annotated genes following RNA sequencing and this was analysed against the genome wide annotation for zebrafish available on the 'Bioconductor' website ('org.Dr.eg.db'), using the Ensembl ID as gene identifier. Results were visualised with 'ridge plot' function in R and GraphPad Prism. The association of differentially expressed genes with human diseases was explored using the DisGeNET database. The human orthologs of the DEGs were identified using BioMart and the R package 'disgenet2r' was used for the analysis. Enrichment results were visualised in GraphPad Prism. The graphics were produced using R, GraphPad Prism, and the Venn diagram tool on the Bioinformatics and Evolutional Genomics site (www.bioinformatics.psb.urgent.be/webtools).

For the analysis of the protein–protein interaction involved in the enriched biological processes identified by GSEA, we used the STRING online tool. The sets of genes identified by GSEA were uploaded onto STRING, the analysis commands were set so that the colour of the nodes was dictated sub-processes served by the respective proteins and the connecting lines between the nodes indicated the confidence or the strength of data supporting the interaction. GO and GSEA analysis identified groups of genes involved in biological processes that were dysregulated by the *cyp21a2* mutation; however, they do not offer

information about the interactions between the proteins which are the products of these genes. The use of STRING provided an analysis of the likely physical interactions between proteins that are co-regulated in mutant zebrafish, which we perceived as an additional confirmation that hypocortisolism caused by 21-hydroxylase deficiency induces gene dysregulations that act synergistically at the molecular level, with an impact on whole processes and pathways. STRING also allowed the visualisation of groups of genes involved in more specific sub-processes or molecular pathways.

## Statistical analysis

Statistical analysis was conducted using GraphPad Prism (GraphPad Software, San Diego, USA). Because of the small number of samples ($n$=6), data normality was assessed using the Shapiro–Wilk test. Normally distributed data were analysed using unpaired $t$-tests, with the Welch's correction for unequal variances. Non-normally distributed data was compared between groups using the Mann–Whitney $U$-test. Statistical significance was considered a $P$-value below 0.05. Graphics were produced using GraphPad Prism and Adobe Illustrator (Adobe Inc., USA).

### Acknowledgements

Parts of the results and discussion in the paper are reproduced from the PhD thesis of Irina A. Bacila (University of Sheffield, 2023). We thank the technical support and aquarium staff for the Zebrafish Facility at the Bateson Centre.

### Competing interests

The authors declare no competing or financial interests.

### Author contributions

Conceptualization: I.B., V.T.C., N.K.; Data curation: I.B., L.O.; Formal analysis: I.B., L.O.; Funding acquisition: I.B., N.K.; Investigation: I.B., L.O., N.L., K.-H.S.; Methodology: I.B., L.O., N.L., K.-H.S.; Project administration: V.T.C.; Resources: I.B., L.O., N.L., K.-H.S., V.T.C., N.K.; Software: I.B.; Supervision: V.T.C., N.K.; Validation: L.O.; Visualization: I.B., L.O.; Writing – original draft: I.B.; Writing – review & editing: I.B., L.O., N.L., K.-H.S., V.C., N.K.

### Funding

This research was funded by a Research Grant to N.K. from the Deutsche Forschungsgemeinschaft (KR 3363/3-1), and by an Early Career Grant to I.B. from The Society for Endocrinology. Open Access funding provided by The University of Sheffield. Deposited in PMC for immediate release.

### Data and resource availability

Relevant data and resources can be found within the article and its supplementary information. The RNAsequencing data, specifically, raw and normalized gene counts, are provided as Supplementary file 1. The raw RNA sequencing data is available at: https://dataview.ncbi.nlm.nih.gov/object/PRJNA1295418?reviewer=1grl5v1qcnlre0fspbfdh3iorc.

### Peer review history

The peer review history is available online at https://journals.biologists.com/bio/lookup/doi/10.1242/bio.061977.reviewer-comments.pdf.

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
