## [Peer Review File · Biology Open]

Steroid 21-hydroxylase deficiency dysregulates essential molecular pathways of metabolism and energy provision

Irina A. Bacila, Lara Oberski, Nan Li, Karl-Heinz Storbeck, Vincent Cunliffe and Nils Krone
DOI: 10.1242/bio.061977

Editor: Daniel Gorelick

Review timeline

Original submission:	6 March 2025
Editorial decision:	24 April 2025
First revision received:	25 June 2025
Editorial decision:	26 June 2025
Second revision received:	24 July 2025
Accepted:	1 August 2025

Original submission

First decision letter

MS ID#: bio.061977

MS Title: Steroid 21-hydroxylase deficiency dysregulates essential molecular pathways of metabolism and energy provision

Authors: Irina A Bacila, Lara Oberski, Nan Li, Karl-Heinz Storbeck, Vincent Cunliffe and Nils Krone

Thanks for submitting your manuscript to Biology Open. I appreciate the opportunity to read and evaluate your work.

The reviewer reports are shown at the bottom of this email or can be accessed, together with a copy of this decision letter, by going to:

As you will see, the reviewers gave favourable reports, but raised some points that will require amendments to your manuscript. I hope that you will be able to carry these out, because we would like to be able to accept your paper.

Many of the reviewer comments can be addressed without additional experiments. If there are any cases where you think a reviewer has proposed an experiment that is unrealistic, please contact me to discuss before you submit a revised version of the manuscript. For reviewer #1, minor point #5, if you choose not to perform more rigorous blood glucose measurements, please do modify the text (and temper the conclusions, if needed) to note the limitations of sampling blood glucose over a 3 hour window following eating. For reviewer #2, comment 1, you do not need to perform any additional experiments to determine whether female weight fluctuates due to the reproductive cycle and oocytes numbers.

Please make sure that RNAseq data is included both as an analyzed Excel table in supplemental data (reviewer #1, major comment #3) and that the raw RNAseq data are deposited in a publicly available database, such as NCBI GEO (my editorial comment, we want the data to be as transparent as possible to enable other labs to analyze following publication).

I'm very much looking forward to seeing a revised version of your interesting manuscript, which I think will be useful for the field.

Reviewer 1

Comments for the author

Bacila et al., investigated the effect of a loss of function of *cyp21a2* in zebrafish on the metabolism. They first characterized the *cyp21a2*^{-/-} embryos and adult zebrafish phenotypically by dosing several steroids, and assessing their size, weight, and fat deposition. They then performed RNA sequencing on wild-type and mutant zebrafish larvae and the livers of wild-type and mutant adult zebrafish to describe the perturbations caused by a lack of glucocorticoids.

This study tackles interesting biological questions: what the consequences of a deficiency in glucocorticoid production on the metabolism are and how can we separate which symptoms of CAH are caused by the deficiency in glucocorticoid and which ones arise following the treatment with glucocorticoids. The model chosen is appropriate to answer the questions, the results are described in details and the statistical analysis has been correctly performed. Therefore, I recommend accepting this article if the following points are addressed:

Major points:

1. The introduction highlights the need for a better characterisation of the consequences of glucocorticoid deficiency and the appropriateness of the zebrafish model to answer the question. However, a section about the exact role of *cyp21a2* both in zebrafish and human and its involvement (or lack thereof) in congenital adrenal hyperplasia is necessary to understand properly why the authors used the *cyp21a2*^{-/-} line to conduct the study.
2. The manuscript would benefit from a more organised description of the transcriptomics experiment results. One could organize it thematically with parts describing the altered pathways of interest (lipid metabolism, glucid metabolism, ATP metabolism...) and comparing their dysregulation in each model (larvae, male liver, and female liver). Enrichment analysis for KEGG pathways is also useful to give an overview of altered metabolic pathways and can be performed easily using tools such as: <https://davidbioinformatics.nih.gov/> or: <https://pathview.uncc.edu/>. A summary schematic could be provided in a figure to help the reader have an overview of the results.
3. Please provide a supplementary Excel file with the full RNAseq results (for all genes, raw and normalized count, log2FoldChange, p-value and adjusted p-value). Also, please consider uploading the raw data on a public repository (e.g., Gene Expression Omnibus).

Minor points:

1. In the introduction, a reference is needed for the sentence "In contrast with human 21-hydroxylase deficiency, this model has normal androgens, which in zebrafish are solely produced in the gonads" (page 4, lines 9-10). Same reference is needed in the discussion, page 17, lines 30-33.
2. Figure 1A is a bit difficult to understand. Perhaps it would be clearer to separate the schematic of the pathway and the plots showing the results of the whole-body steroid measurements.
3. In the results section, the paragraph page 5, lines 27-38 should be written more concisely to be easier to read.
4. The images showing increased fat deposition are clear enough for zebrafish-experienced readers but might not be for someone who never used this model before. If possible, please consider showing sections of the full body (at the brain and intestine levels) stained for lipids (for example, with Oil Red O).
5. For the blood glucose measurements, the fish were fed at 8:00 and the blood was sampled between 9:00 and 12:00. This could explain the high variability of the measure blood glucose, which is typically assessed under fasting condition, and hide a significant difference between the wild-type and mutant fish. If it is not feasible to repeat the experiment, please state it clearly in the results section (page 6, lines 42-48).
6. In the results section, the paragraph ranging from page 7 line 34 to page 8 line 8 should be moved to the material and method section. It is completely normal to have 50-60 % of duplication in RNAseq experiments and a justification is not necessary for this point.
7. In Figure 7, one of the diagrams described in the figure legend seems to be missing.
8. In the material and method section, it would be useful to know if the *cyp21a2* line shows a bias in sex-ratio.
9. In the material and method section, page 25 line 17, it is stated "Sequencing was both paired and single end." This is not possible. Above, it is indicated that the sequencing was paired end for both larvae and adult livers. Please remove this sentence.

10. In the material and method section, page 25 line 39-41, `quantMode GeneCounts` is not utilized to perform differential gene expression analysis, but to count the number of reads per gene.
11. In the material and method section, page 26 line 30-31, I think the authors mean "enriched biological processes" instead of "overexpressed biological processes".
12. In the material and method section, page 26 line 41-43, there is an incomplete sentence.

Reviewer 2

Comments for the author

The manuscript describes an entirely novel Zebrafish model of congenital adrenal hyperplasia (CAH) due to 21-hydroxylase deficiency (`cyp21a2-/-`). This is an important advance since the causes of CAH in the human are well known, but the mechanisms involved in associated comorbidities that are prevalent in patients with CAH due to dysregulation of different metabolic pathways are poorly understood. The authors found that the mutant Zebrafish shared many phenotypic anomalies with human CAH (growth and body fat anomalies) and exhibit changes in metabolic pathways indicating that this is a useful model to study the mechanisms involved and, the long term consequences of, human inherited glucocorticoid deficiency. The strength of this model is the normal production of androgens, which in zebrafish are produced in the gonads. This means that changes in metabolic activity in the mutant fish the is caused by GC deficiency without the influence of androgens. Although the data are mainly descriptive, the study does represent a major step forward to understand the physiology of glucocorticoid deficiency.

Comments -

- (i) In mutant fish the female weight fluctuates - the authors link this to the reproductive cycle and oocytes numbers - can the authors suggest experiments that can control for this?
- (ii) The discussion does not include comparisons with equivalent mouse models of 21-hydroxylase deficiency. This should be commented upon.
- (iii) The authors did not observe a difference in the blood glucose levels between the WT and mutant ZF - can they comment on this finding?
- (iv) The authors found significant changes in ribosomal, ribonucleoprotein, and RNA-related processes in the transcriptome analysis - can the authors relate these transcriptomic changes to the phenotypic changes? Is there evidence (from the human for example) of changes in these processes that mimic the ZF phenotypes?
- (v) Did the `cyp21a2` mutant show any differences in the sex ratio? (PMID 21949597)

Reviewer's Responses to Questions

Experimental quality

Does each figure have the proper controls?

If 'No', please indicate reasons in Comments for Author box below.

Reviewer #1:

- Yes

Reviewer #2:

- No

Were the data analyzed using appropriate statistical tests?

If 'No', please indicate reasons in Comments for Author box below.

Reviewer #1:

- Yes

Reviewer #2:

- Yes

Reproducibility

Were experiments performed using adequate number of biological replicates?
If 'No', please indicate reasons in Comments for Author box below.

Reviewer #1:

- Yes

Reviewer #2:

- No

Does the methods section provide sufficient detail to permit reproducibility?
If 'No', please indicate reasons in Comments for Author box below.

Reviewer #1:

- Yes

Reviewer #2:

- Yes

Completeness

Are the manuscript's conclusions supported by the data?
If 'No', please indicate reasons in Comments for Author box below.

Reviewer #1:

- Yes

Reviewer #2:

- No

Scholarship

Do the authors cite and discuss the merits of data that would argue for and against their conclusion?
If 'No', please indicate reasons in Comments for Author box below.

Reviewer #1:

- Yes

Reviewer #2:

- No

Does the manuscript title & abstract accurately reflect the contents of the manuscript, without hyperbole?

If 'No', please indicate reasons in Comments for Author box below.

Reviewer #1:

- Yes

Reviewer #2:

- No

First revision

Author response to reviewers' comments

Reviewer 1: Bacila et al., investigated the effect of a loss of function of *cyp21a2* in zebrafish on the metabolism. They first characterized the *cyp21a2*^{-/-} embryos and adult zebrafish phenotypically by dosing several steroids, and assessing their size, weight, and fat deposition. They then performed RNA sequencing on wild-type and mutant zebrafish larvae and the livers of wild-type and mutant adult zebrafish to describe the perturbations caused by a lack of glucocorticoids.

This study tackles interesting biological questions: what the consequences of a deficiency in glucocorticoid production on the metabolism are and how can we separate which symptoms of CAH are caused by the deficiency in glucocorticoid and which ones arise following the treatment with glucocorticoids. The model chosen is appropriate to answer the questions, the results are described in details and the statistical analysis has been correctly performed. Therefore, I recommend accepting this article if the following points are addressed:

Major points:

1. The introduction highlights the need for a better characterisation of the consequences of glucocorticoid deficiency and the appropriateness of the zebrafish model to answer the question. However, a section about the exact role of *cyp21a2* both in zebrafish and human and its involvement (or lack thereof) in congenital adrenal hyperplasia is necessary to understand properly why the authors used the *cyp21a2*^{-/-} line to conduct the study.

Response: We thank the reviewer for this suggestion. We have added an additional section to the introduction as recommended.

2. The manuscript would benefit from a more organised description of the transcriptomics experiment results. One could organize it thematically with parts describing the altered pathways of interest (lipid metabolism, glucid metabolism, ATP metabolism...) and comparing their dysregulation in each model (larvae, male liver, and female liver). Enrichment analysis for KEGG pathways is also useful to give an overview of altered metabolic pathways and can be performed easily using tools such as: <https://davidbioinformatics.nih.gov/> or: <https://pathview.uncc.edu/>. A summary schematic could be provided in a figure to help the reader have an overview of the results.

Response: We thank the reviewer for this recommendation. We changed the structure of the results section, grouping the results of the gene set enrichment analysis (GSEA) based on the dysregulated biological processes described. We have also added a diagram conveying the summary of the GSEA findings (Figure 7B).

3. Please provide a supplementary Excel file with the full RNAseq results (for all genes, raw and normalized count, log₂FoldChange, p-value and adjusted p-value). Also, please consider uploading the raw data on a public repository (e.g., Gene Expression Omnibus).

Response: We have provided an excel file as supplementary data as recommended. Once the manuscript is accepted for publication, we will also make the raw data available on a public repository.

Minor points:

1. In the introduction, a reference is needed for the sentence "In contrast with human 21-hydroxylase deficiency, this model has normal androgens, which in zebrafish are solely produced in the gonads" (page 4, lines 9-10). Same reference is needed in the discussion, page 17, lines 30-33.

Response: The finding of normal androgen concentrations in our *cyp21a2*^{-/-} model is reported among the results of the present manuscript. We have amended the wording in the introduction to clarify this point.

2. Figure 1A is a bit difficult to understand. Perhaps it would be clearer to separate the schematic of the pathway and the plots showing the results of the whole-body steroid measurements.

Response: We have amended the figure accordingly.

3. In the results section, the paragraph page 5, lines 27-38 should be written more concisely to be easier to read.

Response: We have amended the paragraph as advised.

4. The images showing increased fat deposition are clear enough for zebrafish-experienced readers but might not be for someone who never used this model before. If possible, please consider showing sections of the full body (at the brain and intestine levels) stained for lipids (for example, with Oil Red O).

Response: We did not undertake full body sections with lipid staining as part of our experiments; thus, they will not be available for this publication. However, we have taken note of the reviewer's suggestion and will seek to address it in future projects using this mutant line.

5. For the blood glucose measurements, the fish were fed at 8:00 and the blood was sampled between 9:00 and 12:00. This could explain the high variability of the measure blood glucose, which is typically assessed under fasting condition, and hide a significant difference between the wild-type and mutant fish. If it is not feasible to repeat the experiment, please state it clearly in the results section (page 6, lines 42-48).

Response: We thank the reviewer for this comment. We have included it in the discussion section, as it is not feasible to repeat the experiments.

6. In the results section, the paragraph ranging from page 7 line 34 to page 8 line 8 should be moved to the material and method section. It is completely normal to have 50-60 % of duplication in RNAseq experiments and a justification is not necessary for this point.

Response: We thank the reviewer for this comment. We have amended the manuscript accordingly.

7. In Figure 7, one of the diagrams described in the figure legend seems to be missing.

Response: We thank the reviewer for highlighting this mistake; this has now been corrected.

8. In the material and method section, it would be useful to know if the *cyp21a2* line shows a bias in sex-ratio.

Response: There is no bias in the sex-ratio identified in this mutant line; we have added this clarification to the methodology.

9. In the material and method section, page 25 line 17, it is stated "Sequencing was both paired and single end." This is not possible. Above, it is indicated that the sequencing was paired end for both larvae and adult livers. Please remove this sentence.

Response: We have removed the line from the text.

10. In the material and method section, page 25 line 39-41, `quantMode GeneCounts` is not utilized to perform differential gene expression analysis, but to count the number of reads per gene.

Response: We have corrected the methodology section.

11. In the material and method section, page 26 line 30-31, I think the authors mean "enriched biological processes" instead of "overexpressed biological processes".

Response: We thank the reviewer for this comment. We have amended the manuscript accordingly.

12. In the material and method section, page 26 line 41-43, there is an incomplete sentence.

Response: We thank the reviewer for highlighting this mistake; we have removed it from the text.

Reviewer 2: The manuscript describes an entirely novel Zebrafish model of congenital adrenal hyperplasia (CAH) due to 21-hydroxylase deficiency (*cyp21a2*^{-/-}). This is an important advance since the causes of CAH in the human are well known, but the mechanisms involved in associated comorbidities that are prevalent in patients with CAH due to dysregulation of different metabolic pathways are poorly understood. The authors found that the mutant Zebrafish shared many phenotypic anomalies with human CAH (growth and body fat anomalies) and exhibit changes in metabolic pathways indicating that this is a useful model to study the mechanisms involved and, the long term consequences of, human inherited glucocorticoid deficiency. The strength of this model is the normal production of androgens, which in zebrafish are produced in the gonads. This means that changes in metabolic activity in the mutant fish the is caused by GC deficiency without the influence of androgens. Although the data are mainly descriptive, the study does represent a major step forward to understand the physiology of glucocorticoid deficiency.

Comments

(i) In mutant fish the female weight fluctuates - the authors link this to the reproductive cycle and oocytes numbers - can the authors suggest experiments that can control for this?

Response: Zebrafish are asynchronous spawners, thus there is no possibility to externally identify stages in the reproductive cycle that would help standardise weight measurements. A possible solution for future experiments would be to compare the difference between body weight and the weight of the ovaries collected and measured during dissection. We have added this comment to the discussion section.

(ii) The discussion does not include comparisons with equivalent mouse models of 21-hydroxylase deficiency. This should be commented upon.

Response: We thank the reviewer for this suggestion. We have added a section to the discussion comparing our results to those from murine models of 21-hydroxylase deficiency.

(iii) The authors did not observe a difference in the blood glucose levels between the WT and mutant ZF - can they comment on this finding?

Response: We thank the reviewer for this comment. We have added a comment to the discussion regarding the blood glucose findings.

(iv) The authors found significant changes in ribosomal, ribonucleoprotein, and RNA-related processes in the transcriptome analysis - can the authors relate these transcriptomic changes to the phenotypic changes? Is there evidence (from the human for example) of changes in these processes that mimic the ZF phenotypes?

Response: We have added the following paragraph to the Discussion:

“A significant impact of cortisol deficiency found in *cyp21a2*^{-/-} mutant larvae was the downregulation of ribosome biogenesis and rRNA processing. Ribosomopathies have been described in the past mostly in relation to the increased risk of oncogenesis (PMC6852887). However, they can associate a number of pathologies, including cardiac defects, facial malformations, hearing loss, liver cirrhosis, neurological impairment. We did not identify any phenotypical features to suggest similar effects in our *cyp21a2*^{-/-} fish. Regarding the metabolic effects of impaired ribosomal function, data from Rpl11-deficient zebrafish and RPS19-deficient mouse showed a suppression of the enzymes involved in glycolysis and the biosynthesis of lipids and proteins, while the expression of genes involved in catabolism is upregulated (PMC4582105). This would suggest that the downregulation of ribosome biogenesis identified by our transcriptomic analysis may have a role in the metabolic dysregulations we found in the adult fish livers.”

(v) Did the *cyp21a2* mutant show any differences in the sex ratio? (PMID 21949597)
Response: There is no bias in the sex-ratio identified in this mutant line; we have added this clarification to the methodology.

Second decision letter

MS ID#: bio.061977R1

MS Title: Steroid 21-hydroxylase deficiency dysregulates essential molecular pathways of metabolism and energy provision

Authors: Irina A Bacila, Lara Oberski, Nan Li, Karl-Heinz Storbeck, Vincent Cunliffe and Nils Krone

Thank you for taking the time to revise your manuscript and respond to the reviewer's comments. You have addressed every comment to our satisfaction and I would love to accept and publish your manuscript, pending one minor revision:

Please upload the raw RNAseq results to a public repository, like NCBI GEO. In your response to reviewer comments you stated that "Once the manuscript is accepted for publication, we will also make the raw data available on a public repository." However, this is not satisfactory - you may understandably be concerned about keeping this data private until publication, but once we officially accept the manuscript, we can't require you to make any changes (although I trust you, we've been burned in the past and so we have a uniform policy to be fair to all authors). Fortunately, NCBI GEO (and perhaps other repositories) have a solution for this, allowing you to upload your raw data but keep it private until the manuscript is published. This privacy mode allows you to provide access to reviewers/editors so that they may review the data and confirm it was uploaded appropriately...once this happens, I will happily publish your revised manuscript. For more information, see here: <https://www.ncbi.nlm.nih.gov/geo/info/faq.html#holduntilpublished>

At this stage, we also ask you to ensure your manuscript complies with our formatting guidelines "please see our manuscript preparation guidelines for details. Provided you are able to fully address the referees' comments, we are positive about publication of your paper (we accept over 95% of revision submissions) and therefore hope you won't mind any extra work involved in reformatting your manuscript at this point.

Please ensure that you clearly highlight all changes made in the revised manuscript. Please avoid using 'Tracked changes' in Word files as these are lost in PDF conversion.

I should be grateful if you would also provide a point-by-point response detailing how you have dealt with the points raised by the reviewers in the 'Response to Reviewers' box. Please attend to all of the reviewers' comments. If you do not agree with any of their criticisms or suggestions please explain clearly why this is so.

Second revision

Author response to reviewers' comments

Thank you for taking the time to revise your manuscript and respond to the reviewer's comments. You have addressed every comment to our satisfaction and I would love to accept and publish your manuscript, pending one minor revision:

Please upload the raw RNAseq results to a public repository, like NCBI GEO. In your response to reviewer comments you stated that "Once the manuscript is accepted for publication, we will also make the raw data available on a public repository." However, this is not satisfactory - you may understandably be concerned about keeping this data private until publication, but once we officially accept the manuscript, we can't require you to make any changes (although I trust you, we've been burned in the past and so we have a uniform policy to be fair to all authors). Fortunately, NCBI GEO (and perhaps other repositories) have a solution for this, allowing you to upload your raw data but keep it private until the manuscript is published. This privacy mode allows you to provide access to reviewers/editors so that they may review the data and confirm it was uploaded appropriately...once this happens, I will happily publish your revised manuscript.

Response: Thank you for accepting to publish our manuscript in Biology Open. We have now uploaded our raw RNAseq data to the NCBI Sequence Read Archive (SRA), and it is available for review at: <https://dataview.ncbi.nlm.nih.gov/object/PRJNA1295418?reviewer=1qr15v1qcnlre0fspbfhdh3iorc> in read-only format. It will remain active and reflect all metadata associated with the manuscript until it is published.

Third decision letter

MS ID#: bio.061977R2

MS Title: Steroid 21-hydroxylase deficiency dysregulates essential molecular pathways of metabolism and energy provision

Authors: Irina A Bacila, Lara Oberski, Nan Li, Karl-Heinz Storbeck, Vincent Cunliffe and Nils Krone

Thank you for depositing your RNA-seq data on NCBI GEO. I am happy to tell you that your manuscript has been accepted for publication in Biology Open, pending our standard publication integrity checks. It was accepted on 1st August 2025.